# Spatial-fluxomics provides a subcellular-compartmentalized view of reductive glutamine metabolism in cancer cells

Won Dong Lee [ID] [1], Dzmitry Mukha[1], Elina Aizenshtein[2] & Tomer Shlomi[1,2,3]

The inability to inspect metabolic activities within subcellular compartments has been a major barrier to our understanding of eukaryotic cell metabolism. Here, we describe a spatial-fluxomics approach for inferring metabolic fluxes in mitochondria and cytosol under physiological conditions, combining isotope tracing, rapid subcellular fractionation, LC-MS-based metabolomics, computational deconvolution, and metabolic network modeling. Applied to study reductive glutamine metabolism in cancer cells, shown to mediate fatty acid biosynthesis under hypoxia and defective mitochondria, we find a previously unappreciated role of reductive IDH1 as the sole net contributor of carbons to fatty acid biosynthesis under standard normoxic conditions in HeLa cells. In murine cells with defective SDH, we find that reductive biosynthesis of citrate in mitochondria is followed by a reversed CS activity, suggesting a new route for supporting pyrimidine biosynthesis. We expect this spatial-fluxomics approach to be a highly useful tool for elucidating the role of metabolic dysfunction in human disease.

[1] Faculty of Biology, Technion, 32000 Haifa, Israel. [2] Lokey Center for Life Science and Engineering, Technion, 32000 Haifa, Israel. [3] Faculty of Computer Science, Technion, 32000 Haifa, Israel. Correspondence and requests for materials should be addressed to T.S. (email: tomersh@cs.technion.ac.il)

Subcellular compartmentalization of metabolic activities is a defining hallmark of eukaryotic cells. Distinct pools of metabolic substrates and enzymes provide cells with flexibility in adjusting their metabolism to satisfy intrinsic demands and respond to external perturbations[1]. Accumulating evidence reveals that the rewiring of metabolic fluxes across organelles supports tumor cell survival and growth[2,3]. For instance, cytosolic one carbon flux can compensate for a loss of the mitochondrial folate pathway[4], and reversed malate-aspartate shuttle across mitochondria and cytosol supports tumor growth upon electron transport chain (ETC) deficiency[5]. Elucidating how metabolic reactions are reprogrammed across organelles is crucial for understanding disease pathologies in eukaryotic cells.

A difficulty in observing metabolic fluxes within distinct subcellular compartments has been a major barrier to our understanding of mammalian cell metabolism[6]. The most direct approach for inferring metabolic flux on a whole-cell level is feeding cells with isotopically labeled nutrients, measuring the isotopic labeling of intracellular metabolites, and computationally inferring flux via Metabolic Flux Analysis (MFA)[7,8]. To estimate compartment-specific fluxes, isotope tracing has been typically applied on purified organelles, though this may suffer from inspecting metabolic flux under non-physiological conditions[9–11]. Alternative approaches such as applying particular isotope tracers[1,2,12], utilizing reporter metabolites either endogenous[4] or engineered[2]; and simulating whole-cell level metabolite isotopic labeling using a compartmentalized flux model[3,13] have provided novel insights to our understanding of compartmentalized metabolism yet may be limited to certain pathways of interest.

A systematic approach for inferring compartmentalized fluxes under physiological conditions requires detecting the isotopic labeling pattern of metabolites in distinct subcellular compartments within intact cells. Reliably measuring metabolite isotopic labeling in mitochondria and cytosol under physiological conditions is highly challenging, considering that conventional cell fractionation approaches typically involve lengthy and perturbative process (e.g., density gradient-based methods taking ~1 h to complete), while the turnover of central metabolic intermediates being in the order of few seconds to minutes[14,15]. Various techniques were proposed for measuring compartment-specific metabolite levels by rapid cell fractionation and quenching of metabolism, including digitonin-based selective permeabilization[16], non-aqueous fractionation (NAF)[17], silicon oil separation[18], high-pressure filtration[19], and recently via immunocapture of epitope-tagged organelles[11,20]. Overall, these studies provided a wealth of information on metabolite levels and key physiological co-factors in distinct subcellular compartments.

Here, we describe a spatial-fluxomics approach for quantifying metabolic fluxes specifically in mitochondria and cytosol, performing isotope tracing in intact cells followed by rapid subcellular fractionation and LC-MS-based metabolomics analysis. Using an optimized fractionation method, we achieve subcellular fractionation and quenching of metabolism within 25 s. Computational deconvolution with metabolic and thermodynamic modeling enables the inference of compartment-specific metabolic fluxes. We apply the spatial-fluxomics method to investigate mitochondrial and cytosolic fluxes involved in reductive glutamine metabolism, mediating fatty acid biosynthesis under hypoxia[21], in cells with defective mitochondria[22], and in anchorage-independent growth[3]. Specifically, under these conditions, acetyl-CoA (a precursor for fatty acid biosynthesis) was shown to be primarily synthesized via reductive isocitrate dehydrogenase (IDH), producing citrate from glutamine-derived α-ketoglutarate (αKG), which is cleaved by ATP citrate lyase (ACLY) to produce cytosolic oxaloacetate (OAA) and acetyl-CoA. Surprisingly, we find that reductive glutamine metabolism is, in fact, the major producer of cytosolic citrate (rather than glucose oxidation) to support fatty acid biosynthesis also under standard normoxic conditions in HeLa cells (in contrast to the canonical view where cytosolic citrate is derived via a transport of mitochondrial citrate, which is produced by glucose oxidation through the TCA cycle). We further reveal that while the relative contribution of reductive glutamine metabolism to fatty acid production (versus that of glucose oxidation) increases under hypoxia, the total reductive flux drops compared to that of normoxia. Analyzing metabolic reprogramming in succinate dehydrogenase (SDH) deficiency, the spatial-fluxomics approach revealed that SDH-deficient tumors reverse the citrate synthase (CS) flux to produce OAA in mitochondria to support pyrimidine biosynthesis (instead of through the canonical pathway via shuttling of the mitochondrial citrate to cytosol). This demonstrates the potential usage of the spatial-fluxomics method in identifying novel pathways that may be therapeutically targeted.

## Results

**Quantifying compartmentalized metabolite levels.** We optimized a protocol for rapid separation of mitochondrial and cytosolic metabolites utilizing digitonin and centrifugation, followed by rapid quenching of metabolism within 25 s[16]. Applying this method to HeLa cells and tracking a mitochondrial marker (MitoTracker Deep Red) with confocal microscopy, we confirmed that mitochondrial membrane remains intact post fractionation (Fig. 1a). To quantitatively assess the purity of the two subcellular fractions, we measured the concentration of mitochondrial (citrate synthase; CS) and cytosolic (glyceraldehyde phosphate dehydrogenase; GAPDH) reporter proteins; and mitochondrial (tetramethylrhodamine methyl ester; TMRM; sequestered by active mitochondria) and cytosolic (glucose-6-phosphate) markers (Fig. 1b–d and Supplementary Fig. 1). We find that the derived mitochondrial and cytosolic fractions are ~90% pure, with ~10% cross-contamination between the two fractions (i.e., ~10% of the cytosolic content appearing in the mitochondrial fraction and vice versa). Notably, while nuclei remain within the mitochondrial fraction (Fig. 1a), nuclear metabolites are expected to diffuse into the cytosolic fraction through nuclear pore complexes (NPCs; freely permeable to small molecules 5000 Da or less)[23,24].

We utilized LC-MS to quantify the relative abundance of 42 metabolites in the mitochondrial and cytosolic fractions out of the total cellular pool (employing isotope ratio[25] to accurately compare concentrations measured in the two fractions; Methods; Supplementary Data 1). To assess whether metabolism is perturbed throughout the fractionation procedure, we compared the sum of metabolite pool sizes measured in the two fractions to that measured on a whole-cell level (where physiological pool sizes are obtained by immediately quenching the metabolism of live cells with a cold solvent). We found that for ~90% of the metabolites, the sum of the measured pool sizes in the mitochondrial and cytosolic fractions was less than 20% off that measured in whole-cells (Fig. 1e and Supplementary Data 1). For example, the sum of citrate and malate pool sizes in the two fractions perfectly matched to that of the whole-cell measurement; and for alpha-ketoglutarate, the sum of pool sizes deviated less than 13% of the whole-cell measurement. The overall deviation between the sum of pool sizes in the two fractions and the whole-cell measurements was significantly smaller than that expected by chance (Wilcoxon $p$-value = 0.005; comparing the distribution of deviations from the whole-cell measurements with that obtained when randomly pairing metabolite pool size measurements in the mitochondrial and cytosolic fractions). Notably, delaying the metabolite extraction from the

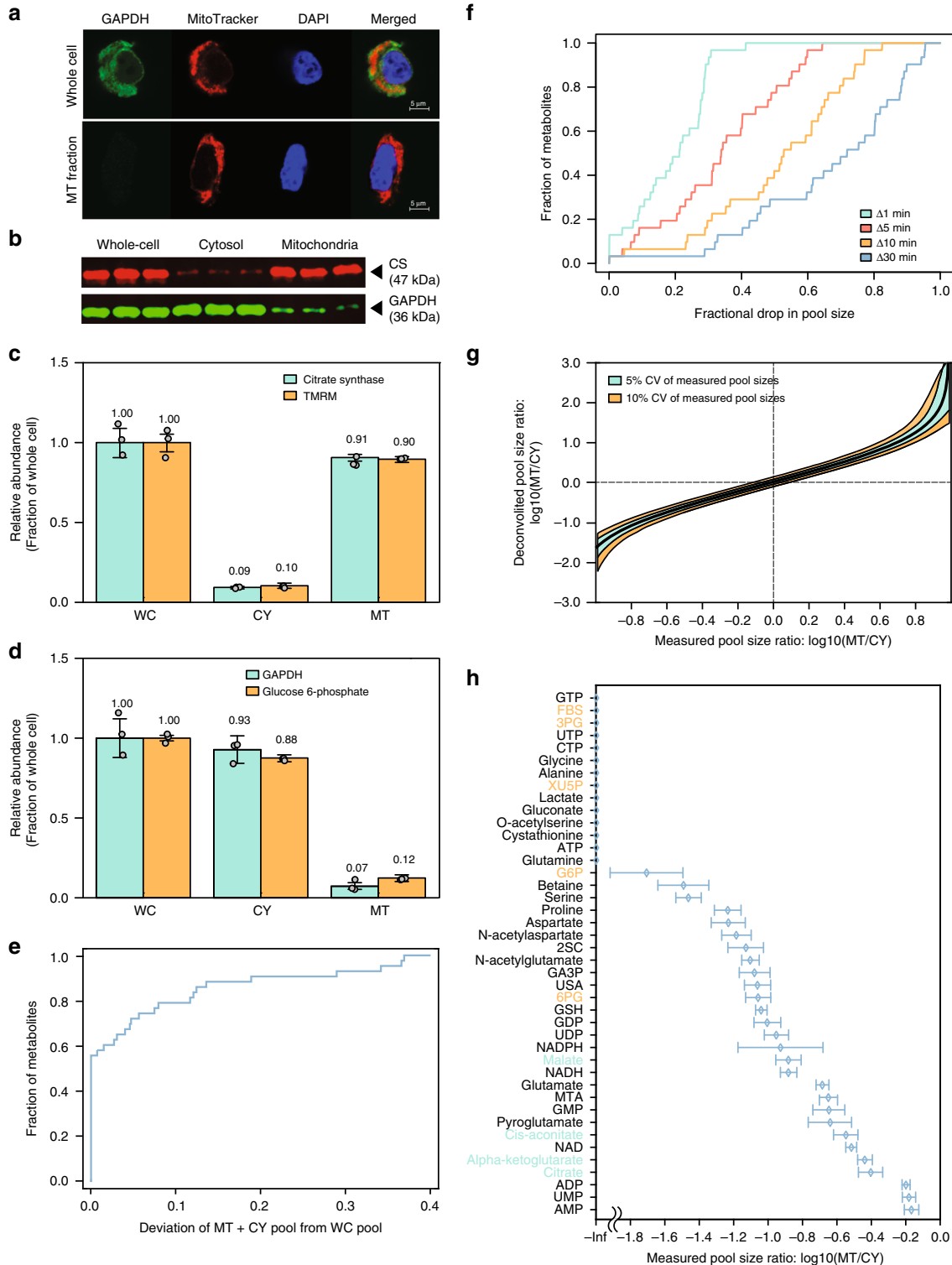

mitochondrial fraction by a few minutes resulted in diffusion of metabolites from the mitochondrial matrix (e.g. ~50% drop in the pool size of ~30% of the metabolites when the method is lengthened by 5 min; with mitochondria kept in PBS), emphasizing the need for rapid fractionation and quenching of mitochondrial metabolism (Fig. 1f).

A computational deconvolution approach was utilized to overcome the impurity of the derived subcellular fractions and to accurately quantify metabolite pool sizes in mitochondria and cytosol (Methods; Fig. 1g and Supplementary Data 1). Expectedly,

the inferred relative pool sizes of glycolytic and pentose phosphate pathway metabolites were significantly lower in mitochondria (Wilcoxon $p$-value = 0.018; mitochondrial pool size is ~2% that of cytosol, on average) (Fig. 1h). The relative pool sizes of tricarboxylic acid (TCA) cycle metabolites (citrate, aconitate, αKG, and malate) were significantly higher in mitochondria (Wilcoxon $p$-value = 0.016), with a mean relative pool size of ~30%. Considering that the total volume of mitochondria being ~10-fold lower than that of cytosol (volume estimations for total mitochondria over whole-cell range between 5–20% in different

**Fig. 1** An approach for quantifying mitochondrial and cytosolic metabolite levels. **a** Rapid cell fractionation effectively removes cytosolic components (green) without damaging mitochondrial membrane. Mitochondria retain MitoTracker (red) after digitonin-based cell fractionation. While nucleus (blue) remains intact within the mitochondrial fraction, it should not bias the mitochondrial metabolite pool size measurements due to the free diffusion of small molecules through nuclear pore complexes into the cytosolic fraction. **b** The purity of the mitochondria and cytosol-enriched fractions in terms of the expression of mitochondrial (citrate synthase; CS, red) and cytosolic marker proteins (glyceraldehyde-phosphate dehydrogenase; GAPDH, green) via quantitative western blot (three biological replicates are shown). **c–d** The extent of cross-contamination between the mitochondrial and cytosolic fractions based on mitochondrial and cytosolic protein (western blot, green) and small-molecule markers (LC-MS, orange). Tetramethylrhodamine methyl ester (TMRM) was introduced as a mitochondria-specific small-molecule marker. **e** A comparison of the sum of measured metabolite pools in the two subcellular fractions to that measured in whole-cell extracts; a cumulative distribution shows the number of metabolites (y-axis) for which the sum of pool sizes in the two fractions deviates by different extent from the whole-cell measurements (x-axis). **f** Delay in the quenching of metabolism in the mitochondrial fraction (by 1, 5, 10, and 30 min) leads to reduced pool sizes due to the diffusion of small molecules out of mitochondria; a cumulative distribution shows the number of metabolites (y-axis) for which the fractional pool size in the mitochondrial fraction drops by different extent (x-axis). **g** The deconvolution function used to infer metabolite pool size ratio in mitochondria versus cytosol (y-axis) given the measured pool size ratio (x-axis), considering the cross-contamination between the two fractions. Confidence intervals of the deconvoluted pool size rations are shown considering 5% (orange) and 10% (green) coefficient of variance (CV) in the measured pool sizes in the mitochondrial and cytosolic fractions. **h** The ratio of metabolite pool sizes in mitochondria versus cytosol for 42 metabolites in HeLa cells under standard normoxic condition; glycolytic and pentose phosphate pathway metabolites (orange) and TCA cycle metabolites (green). Data are mean ± SD, $n = 3$ independent biological replicates

human cell lines[26]; and ~7% for HeLa[27]), the concentrations of TCA cycle intermediates in mitochondria are ~3-fold higher on average in mitochondria than in cytosol. The higher concentration of these metabolites in mitochondria is consistent with previous measurements in isolated rat hepatocytes using digitonin-silicon oil method[18], supporting the reliability of the pool size measurements performed following the rapid cell fractionation.

**Compartmentalized isotope tracing in mitochondria and cytosol.** To probe compartmentalized metabolic fluxes, we fed HeLa cells with [U-$^{13}$C]-glucose or [U-$^{13}$C]-glutamine for different time periods, followed by rapid fractionation and quenching of metabolism. LC-MS was applied to measure the mass-isotopomer distribution (MID) of metabolites in each subcellular fraction (Fig. 2a, b; Methods)—i.e., the fraction of metabolite pools having zero, one, two, etc labeled carbons (denoted as m + 0, m + 1, m + 2, respectively). To account for the impurity of the mitochondrial and cytosolic fractions, we applied a deconvolution approach for inferring the mitochondrial and cytosolic MIDs (Methods; considering that a metabolite with a substantially larger pool size in one compartment will strongly contaminate the measured MIDs in the other).

Tracing isotopic glucose, we observed rapid labeling of citrate m + 2 in mitochondria followed by slower labeling in the cytosol, consistent with the canonical view of citrate biosynthesis in mitochondria via CS, followed by its shuttling to the cytosol (Fig. 2c). The transport of mitochondrial citrate to cytosol was also observable when feeding isotopic glutamine where rapid labeling of mitochondrial citrate m + 4 was followed by slower labeling kinetics in the cytosol (Fig. 2d). Tracing isotopic glutamine, we observed faster labeling of intracellular glutamine m + 5 in cytosol before its entry into mitochondria (Fig. 2e). Glutamate m + 5 was labeled faster in mitochondria, suggesting that mitochondrial glutaminase is the prime route for glutamate production from glutamine (Fig. 2f), in accordance with the known localization of this enzymatic activity in cancer cells[22].

Reductive glutamine metabolism through reverse IDH activity was evident by the m + 5 labeling of citrate in mitochondria and cytosol, when feeding [U-$^{13}$C]-glutamine (Fig. 3a, b). The faster labeling of mitochondrial citrate m + 5 versus cytosolic citrate m + 5 indicates a reduction of glutamine-derived αKG by the mitochondrial IDH (fractional labeling of citrate m + 5 being ~6% in mitochondria versus ~3.5% in cytosol after 3 h). On the other hand, the ratio between citrate m + 5 and m + 4 after 3-h feeding of isotopic glutamine was ~60% higher in cytosol than in mitochondria, indicating the presence of reductive IDH flux also

in the cytosol (Supplementary Data 3). Below, we employ quantitative flux modeling to infer the exact reductive IDH flux in each compartment.

Malate had markedly higher m + 3 labeling in cytosol than in mitochondria (Fig. 3c, d; ~8% labeled in cytosol versus ~3% in mitochondria after 12 h). A potential source for the excess cytosolic malate m + 3 is malate dehydrogenase, which consumes OAA m + 3 that is produced from citrate m + 5 via cytosolic ACLY (Supplementary Fig. 2). Interestingly, however, the fractional labeling of cytosolic malate m + 3 was consistently higher than that of citrate m + 5 in the cytosol (~4% after 12 h), suggesting that there is another source for cytosolic malate m + 3. Analyzing the positional carbon labeling of cytosolic malate (based on the isotopic labeling pattern of uridine triphosphate; Supplementary Fig. 2) suggests that malate m + 3 is produced from malate m + 4 through an isotopic exchange with $CO_2$ via malic enzyme (ME1) or phosphoenolpyruvate carboxykinase (PCK1) in the cytosol.

To further validate that rapid fractionation accurately captures compartment-specific metabolite labeling kinetics, we compared the measured labeling kinetics of citrate and malate in each compartment to that detected in the culture media (following secretion from cells). Measuring the isotopic labeling pattern of citrate and malate in media 24 h post feeding cells with [U-$^{13}$C]-glutamine, we found a good match between the experimental measurements in media and the expected labeling pattern based on the cytosolic labeling kinetics (considering that labeling patterns in media represent the aggregated labeling of the cytosolic metabolite pools within the 24 h time period; Fig. 3e).

**Quantifying compartmentalized reductive glutamine metabolism.** To infer the actual flux through the various mitochondrial and cytosolic IDH isozymes, we employed computational modeling based on Kinetic Flux Profiling (KFP)[28], utilizing compartment-specific metabolite pool size measurements, thermodynamic constraints, and compartment-specific isotopic labeling data (Methods). We constructed a compartmentalized flux model of citrate metabolism (Supplementary Fig. 3) that consist of distinct pools for citrate and αKG in mitochondria and cytosol, transport of citrate between the two compartments, and the various IDH isozymes: cytosolic IDH (IDH1, NADP-dependent) and mitochondrial IDHs (IDH2, NADP-dependent; IDH3, NAD-dependent). Thermodynamics was used to infer the direction of net flux and ratio between the forward and backward flux through various IDH isozymes[29]. Specifically, absolute measurement of metabolite pool sizes in mitochondria and cytosol

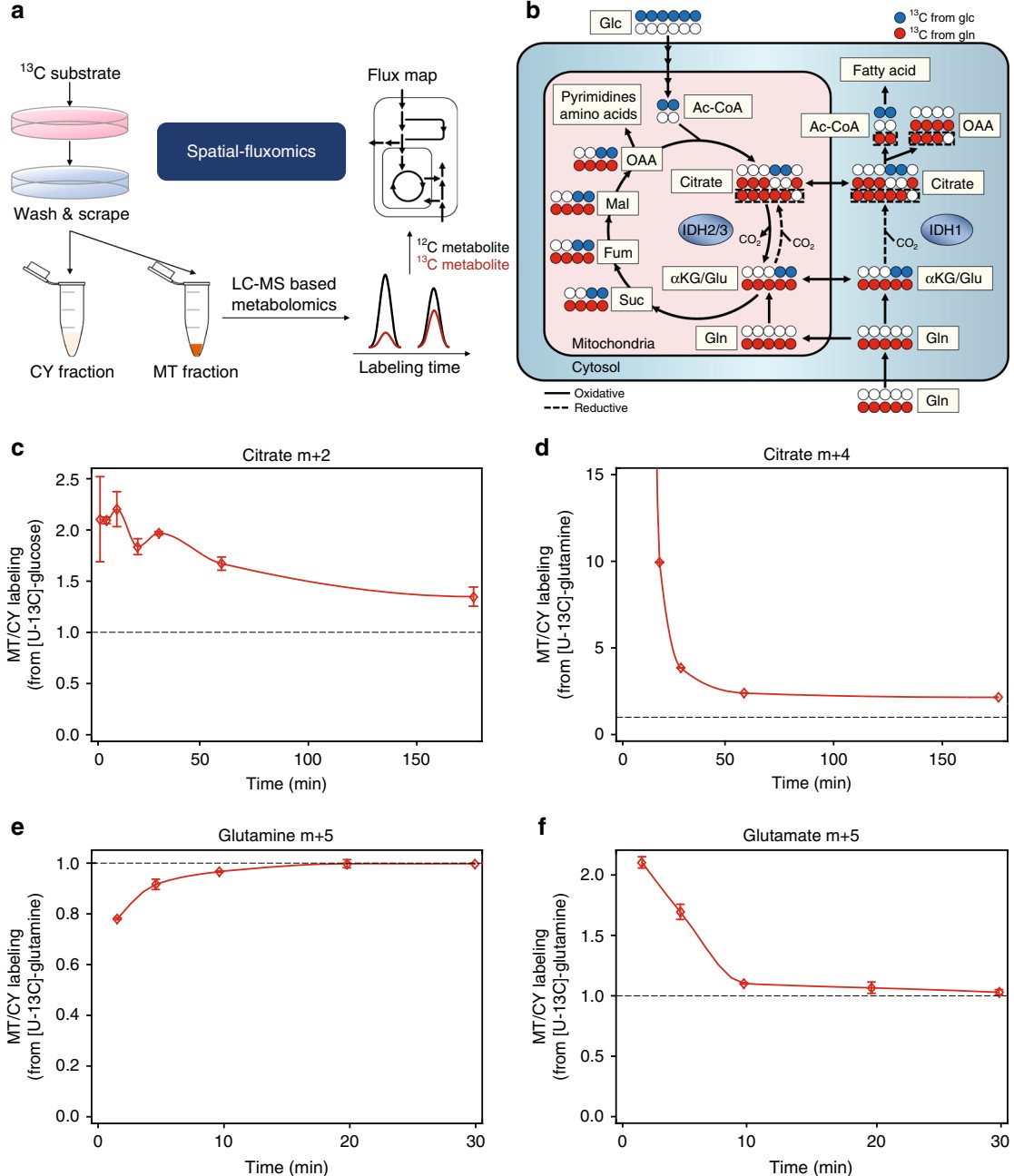

**Fig. 2** Compartment-specific isotope tracing in mitochondria and cytosol. **a** A schematic description of the spatial-fluxomics approach. **b** TCA cycle metabolism and associated reactions involving acetyl-CoA and fatty acid metabolism. **c** Isotopic labeling kinetics of citrate $m + 2$ in mitochondria versus cytosol when feeding HeLa cells with [U-$^{13}$C]-glucose; showing the ratio between the relative abundance of the $m + 2$ form of citrate in the mitochondria (out of the entire mitochondrial pool) divided by the relative abundance of citrate $m + 2$ in cytosol. **d-f** The isotopic labeling kinetics of citrate $m + 4$ (**d**), glutamine $m + 5$ (**e**), and glutamate $m + 5$ (**f**) when feeding [U-$^{13}$C]-glutamine. Data are mean ± SD, $n = 3$ independent biological replicates

enabled the calculation of Gibbs free energy for reactions catalyzed by each IDH isozyme (Fig. 3f and Supplementary Data 2; Methods). Fluxes obtained by this approach provided good agreement with the experimentally observed metabolite isotopic labeling kinetics (Fig. 3g and Supplementary Figs. 4–8). Net and forward/backward fluxes with 95% confidence intervals are included in Supplementary Data 8.

We found that mitochondrial NAD-dependent IDH3 has a net flux in the oxidative direction while NADP-dependent IDH1 and IDH2 have net fluxes in the reductive direction: IDH3 oxidizes citrate in a rate roughly similar to glycolytic flux into TCA cycle via CS (0.48 mM h$^{-1}$). The reductive flux through mitochondrial

IDH2 (~14% of CS flux) and cytosolic IDH1 (~2% of CS flux) were ~8-times and ~58-times lower than that of IDH3, respectively. The NAD-dependent IDH3 was far from chemical equilibrium ($\Delta_r G' < -10$ kJ mol^−1), while NADP-dependent IDHs in both mitochondria and cytosol were potentially close to chemical equilibrium (where the large confidence interval is due to uncertainty in the estimation of the standard Gibbs free energy; Fig. 3f). This suggests a potential large backward flux in the oxidative direction for both IDH1 and IDH2. Notably, the rates of NADPH oxidation by IDH1 and IDH2 were small compared to the total NADPH production rate, which is estimated to be on the order of a few mM h$^{-1}$ (ref. [30]).

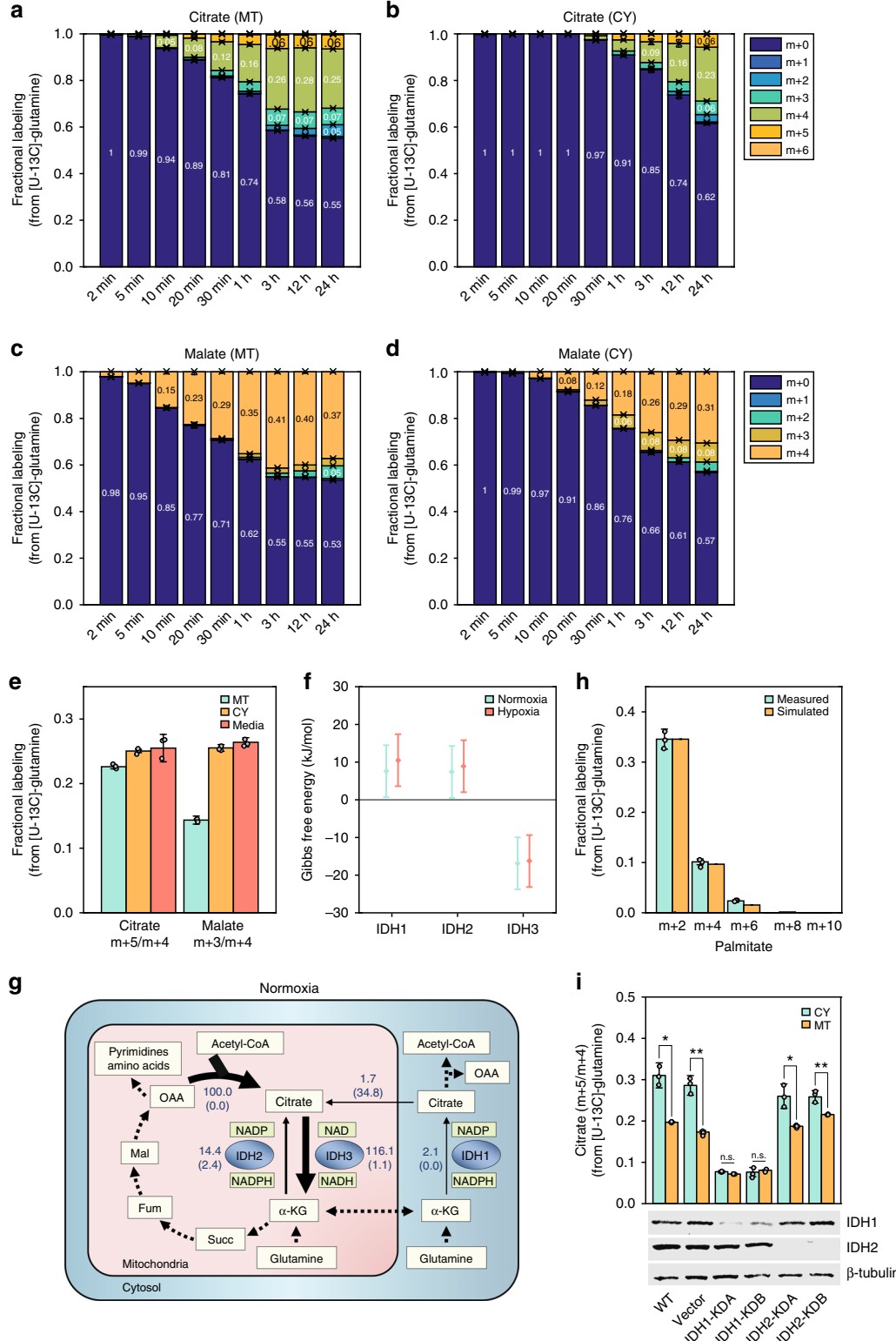

Interestingly, citrate was found to be rapidly transported from mitochondria to cytosol and vice versa, with no net flux along the canonical route from mitochondria to the cytosol (net flux from cytosol to mitochondria is up to ~2% of CS flux; Supplementary Data 8). Shuttling of cytosolic citrate into mitochondria was recently reported to support redox homeostasis during anchorage-independent growth[3], yet we suggest that this also occurs under standard tissue culture conditions. Considering no

net transport of citrate from mitochondria to cytosol, reductive IDH1 is the sole net producer of cytosolic citrate in HeLa cells under normoxia. Notably, whole-cell level measurement of citrate labeling cannot elucidate the important role of IDH1 under normoxia due to the rapid bi-directional transport of citrate across the mitochondrial membrane that dilutes the cytosolic citrate labeling (and hence also the labeling of acetyl-CoA and fatty acids). And indeed, measuring fatty acid labeling 24 h post

**Fig. 3** A quantitative view of mitochondrial and cytosolic fluxes in the TCA cycle and citrate metabolism under normoxia. **a–d** Mass-isotopomer labeling kinetics of citrate in mitochondria (**a**) and cytosol (**b**), and malate in mitochondria (**c**) and cytosol (**d**) when feeding HeLa cells with [U-$^{13}$C]-glutamine under standard normoxic conditions. **e** Measured isotopic labeling ratio in HeLa cells for citrate m + 5 /m + 4 and malate m + 3 /m + 4 in media (red) in comparison to the expected labeling via computational simulation, considering the measured labeling kinetics of citrate and malate in mitochondria (green) and cytosol (orange). **f** Gibbs free energy of mitochondrial (IDH2/3) and cytosolic (IDH1) IDH isozymes (in the oxidative direction) in HeLa cells under normoxia (green) and hypoxia (red). **g** Mitochondrial and cytosolic fluxes, showing percentage from citrate synthase flux (which is 0.48 mM h$^{-1}$). Arrow represents the direction of net flux; number represents net flux in the direction of the arrow and number in parenthesis correspond to the backward flux. Confidence intervals for estimated fluxes are shown in Supplementary Data 8. **h** The measured mass-isotopomer distribution of palmitate when feeding cells with [U-$^{13}$C]-glutamine (green) is consistent with the simulated fit (orange). For the simulation, acetyl-CoA labeling was assumed to follow a binomial distribution with a probability of 7.1% having m + 2 labeling form. **i** Validation of the method based on knock-down of IDH1 or IDH2 genes and following citrate isotopic labeling after feeding cells with [U-$^{13}$C]-glutamine. Upon IDH1 knockdown, the ratios between citrate m + 5 and m + 4 in mitochondria and cytosol are similar, indicating that all reductive glutamine flux occurred in mitochondria (where citrate m + 4 is produced from malate m + 4). Meanwhile, IDH2 knockdown resulted in a higher citrate m + 5 to m + 4 ratio in cytosol, indicating that reductive IDH1 remains active. n.s. not significant. *$P < 0.05$ and **$P < 0.01$ by two-sample $t$-test. Data are mean ± SD, $n = 3$ independent biological replicates

feeding cells with isotopic glutamine and utilizing Isotopomer Spectral Analysis (ISA)[31,32] revealed that only ~7% of cytosolic acetyl-CoA is in the m + 2 form (resulting from reductive IDH flux) (Fig. 3h). The spatial-fluxomics analysis in HeLa cells grown with a physiological concentration of glutamine (0.5 mM) showed qualitatively similar results, with no net transport of mitochondrial citrate to the cytosol (Supplementary Fig. 9 and Supplementary Data 8).

To further validate the method, we knocked down IDH1 or IDH2 and followed citrate isotopic labeling after feeding cells with [U-$^{13}$C]-glutamine (Fig. 3i). Expectedly, knocking down cytosolic IDH1, the ratios between citrate m + 5 and m + 4 in mitochondria and cytosol were similar, indicating that all reductive carboxylation of αKG occurred through the mitochondrial IDH2 (where citrate m + 4 is produced from malate m + 4 via CS). Meanwhile, the knockdown of mitochondrial IDH2 resulted in a higher citrate m + 5 to m + 4 ratio in the cytosol, indicating predominant reductive flux through IDH1 in the cytosol.

**Mitochondrial reductive IDH2 flux drops in hypoxia**. To explore metabolic alterations in hypoxia, we grew HeLa cells under 1% oxygen. When cells were fed with [U-$^{13}$C]-glutamine, we detected a major increase in the fractional labeling of citrate m + 5 in both mitochondria and cytosol, in accordance with previous whole-cell measurements—representing an increase in the relative contribution of reductive glutamine metabolism to fatty acid biosynthesis in hypoxia (Fig. 4a, b)[21]. Analyzing the cellular response to hypoxia in terms of compartment-specific metabolite concentrations, we found a major ~70% drop in citrate level in both mitochondria and cytosol ($t$-test $p$-value < 0.001; Fig. 4c; Supplementary Data 2); while αKG showed a significant 50% drop only in mitochondria ($t$-test $p$-value < 0.001). The resulting 2-fold higher αKG/citrate ratio in cytosol versus that in mitochondria under hypoxia suggests a potential shift in the balance of reductive flux between the two compartments toward higher flux in the cytosol[33].

Computational flux analysis utilizing the compartmentalized metabolic measurements shows that the net glutamine reductive flux does not increase in hypoxia compared to normoxia. In fact, the mitochondrial reductive IDH2 net flux shows a significant >55% drop in hypoxia, while the cytosolic reductive IDH1 flux does not change significantly (Fig. 4d; see 95% confidence interval of net fluxes in Supplementary Data 8). The major increase in fractional citrate m + 5 labeling in both mitochondria and cytosol under hypoxia is hence solely due to a marked ~95% drop in glycolytic flux into the TCA cycle through CS. The latter is in agreement with the known metabolic reprogramming associated with hypoxia-inducible factor 1 (HIF1) where stabilization of HIF1 under hypoxia induces pyruvate dehydrogenase kinase 1

(PDK1) activity, which inhibits pyruvate dehydrogenase (PDH) and hence glycolytic flux into the TCA cycle[34].

**Reversal of CS in Krebs cycle supports growth of SDH-null cells**. Loss-of-function mutations in the SDH (succinate dehydrogenase; respiratory Complex II) subunits are associated with the development of neuroendocrine neoplasm, gastrointestinal stroma, and renal cell carcinoma[35]. SDH-deficient cells compensate for the impaired mitochondrial function by inducing aerobic glycolysis and diverting glucose-derived flux toward TCA cycle anaplerosis (i.e., for the replenishment of TCA cycle intermediates) via pyruvate carboxylase (PC)[36]. Reductive glutamine metabolism is another major source of anaplerotic flux in SDH-deficient cells[22,37]. Consistently, feeding SDH-deficient cells with [U-$^{13}$C]-glutamine, we observed substantial labeling of citrate m + 5 (~87%) and malate m + 3 (~75%) (Supplementary Fig. 10 and Fig. 5a, b).

Measuring changes in compartmentalized metabolite pools upon SDH loss, we detected a significant ~84% ($t$-test $p$-value = 0.002) drop in the citrate pool size specifically in mitochondria, while the cytosolic pool remained similar to that in the wild-type SDH cell line (Fig. 5c and Supplementary Data 2). Together with a ~2-fold increase in αKG level in both compartments, the αKG/citrate ratio in mitochondria increased more than 20-fold, suggesting a potential shift in the balance of reductive flux between the two compartments toward higher flux in mitochondria. Meanwhile, the total cellular pool of NADH increased 4-fold in SDH-deficient cells, with the mitochondrial pool specifically showing a significant ~10-fold increase compared to that of parental SDH-positive cells (Fig. 5c). The accumulation of NADH in mitochondria is in accordance with the defective ETC activity and decreased respiration of SDH-deficient cells. The marked drop in mitochondrial citrate and an increase in mitochondrial NADH in the SDH-deficient cells result in the mitochondrial IDH3 moving closer to chemical equilibrium (Fig. 5d).

Quantitative flux modeling showed that reductive IDH2 flux in mitochondria increases ~10-fold in SDH-defective cells compared to the wild-type cells (resulting in more than 7-fold higher reductive glutamine flux in mitochondria than in cytosol; Fig. 5e; see flux confidence intervals in Supplementary Data 8). Surprisingly, the analysis suggests that most (>70%) of the reductively synthesized citrate in mitochondria flow in the reversed direction through CS, producing OAA in mitochondria to support amino acid and pyrimidine biosynthesis—rather than through the canonical pathway via ACLY in the cytosol. In fact, ACLY not being the sole producer of reductively synthesized OAA (and then of malate) in these cells is readily observable from the compartmentalized isotopic labeling data, as SDH-deficient cells have faster labeling of mitochondrial malate m + 3 than of cytosolic citrate m + 5 (Fig. 5b and Supplementary Fig. 11).

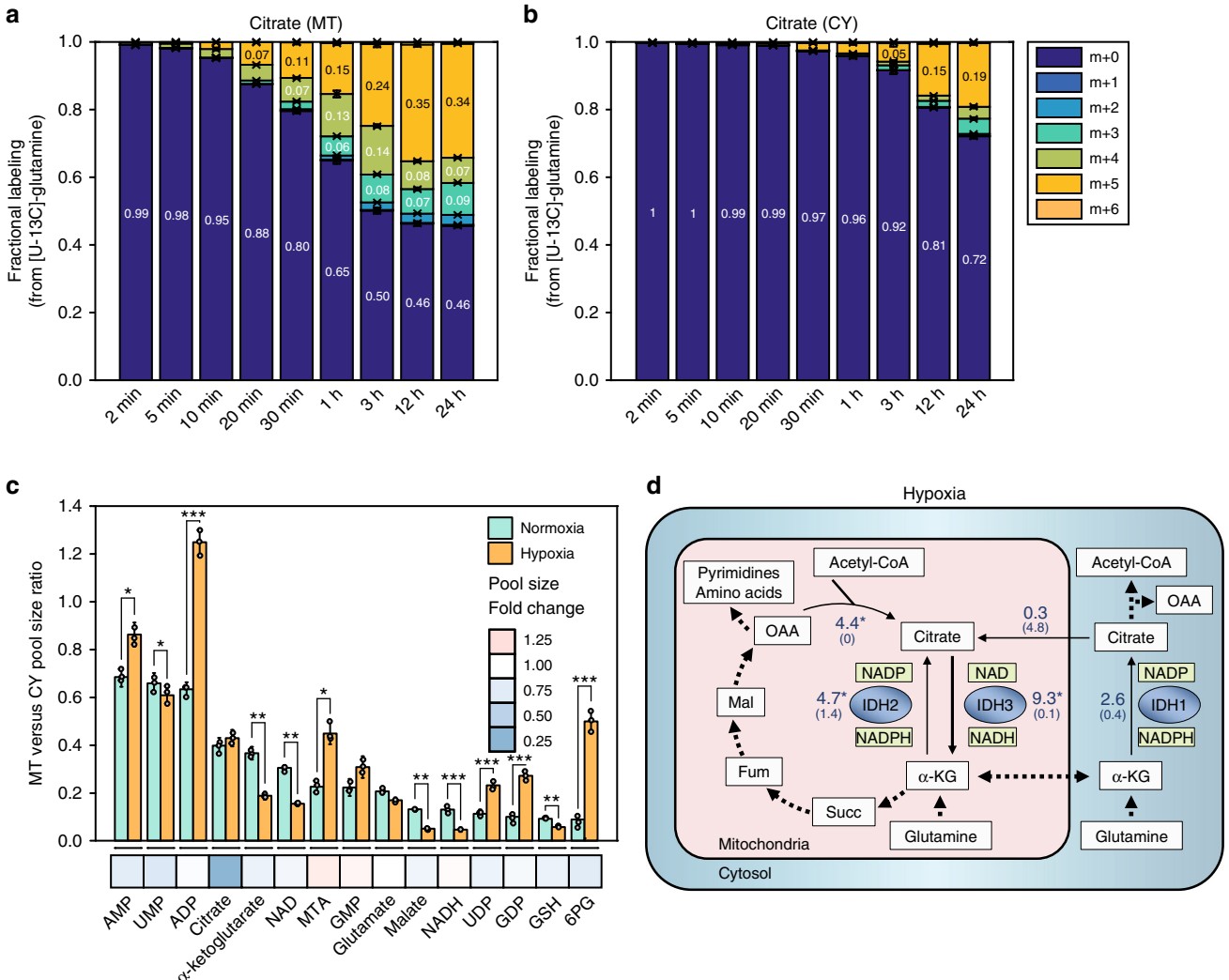

**Fig. 4** Metabolic rewiring of mitochondrial and cytosolic fluxes under hypoxia. **a, b** Mass-isotopomer labeling kinetics of citrate in mitochondria (**a**) and cytosol (**b**) when feeding HeLa cells with [U-13C]-glutamine under hypoxia. **c** Mitochondrial to cytosolic metabolite pool size ratios (y-axis) under normoxia (green) and hypoxia (orange) in HeLa cells; average fold change in whole-cell pool size under normoxia and hypoxia is indicated by color (x-axis). **d** Mitochondrial and cytosolic fluxes in HeLa cells under hypoxia, showing percentages from citrate synthase flux in normoxia. Arrow represents the direction of net flux; number represents net flux in the direction of the arrow and number in parenthesis correspond to the backward flux. Confidence intervals for estimated fluxes shown in Supplementary Data 8. *$P < 0.05$, **$P < 0.01$, and ***$P < 0.001$ by two-sample t-test. Data are mean ± SD, $n = 3$ independent biological replicates

To confirm that SDH-deficient mitochondria catalyze reverse CS flux, we further performed isotope tracing experiments in isolated mitochondria from both SDH-WT and KO cells feeding [U-13C]-citrate (Methods). The depletion of ACLY from the mitochondrial isolate was validated by western blot (Fig. 5f). Reverse flux through CS in isolated mitochondria of SDH-KO cells was evident by the m + 2 labeled acetyl-CoA (~51%) and m + 4 malate (~54%) (Fig. 5g; direct measurement of OAA labeling was impossible via LC-MS due to its instability[38]). Notably, malate m + 4 cannot be generated through oxidative TCA cycle metabolism of citrate in SDH-deficient mitochondria due to the loss of SDH (as evident by no detectable malate m + 4 when feeding SDH-KO cells with [U-13C]-glutamine; Table S6). On the contrary, we did not find evidence of reverse CS flux in SDH-WT mitochondria, as acetyl-CoA m + 2 is non-detectable (though a substantial amount of malate m + 4 is produced via oxidative TCA cycle activity; Fig. 5g). The non-detectable acetyl-CoA m + 2 in SDH-WT mitochondria suggests no residual ACLY activity in the mitochondrial isolates. Indeed, the addition

of ATP and the ACLY inhibitor (BMS-303141) to mitochondrial isolates from SDK-KO cells did not affect the relative abundance of acetyl-CoA m + 2 or malate m + 4 (Fig. 5h). Furthermore, ATP-independent citrate cleavage activity was found in SDH-KO cell lysates (Supplementary Fig. 12b); and *ACLY* silencing had a marginal effect on reductive biosynthesis of malate in SDH-deficient cells measured via [1-13C]-glutamine tracing (Supplementary Fig. 12c–d).

Considering a potential role of reversed CS flux in reductive biosynthesis of amino acids and pyrimidines in SDH-deficient cells, we hypothesized that inhibiting this reversed flux would selectively halt the proliferation of these cells. To test our hypothesis, we treated SDH-positive and negative cells with dichloroacetate (DCA), a PDK inhibitor, which was shown to induce glucose-derived acetyl-CoA production and its oxidation via CS[39]—and is hence expected to hamper net CS flux in the reverse direction and ultimately pyrimidine biosynthesis. Indeed, we found that SDH-null cells display significantly higher sensitivity to DCA treatment (Fig. 5i). DCA treatment inhibited

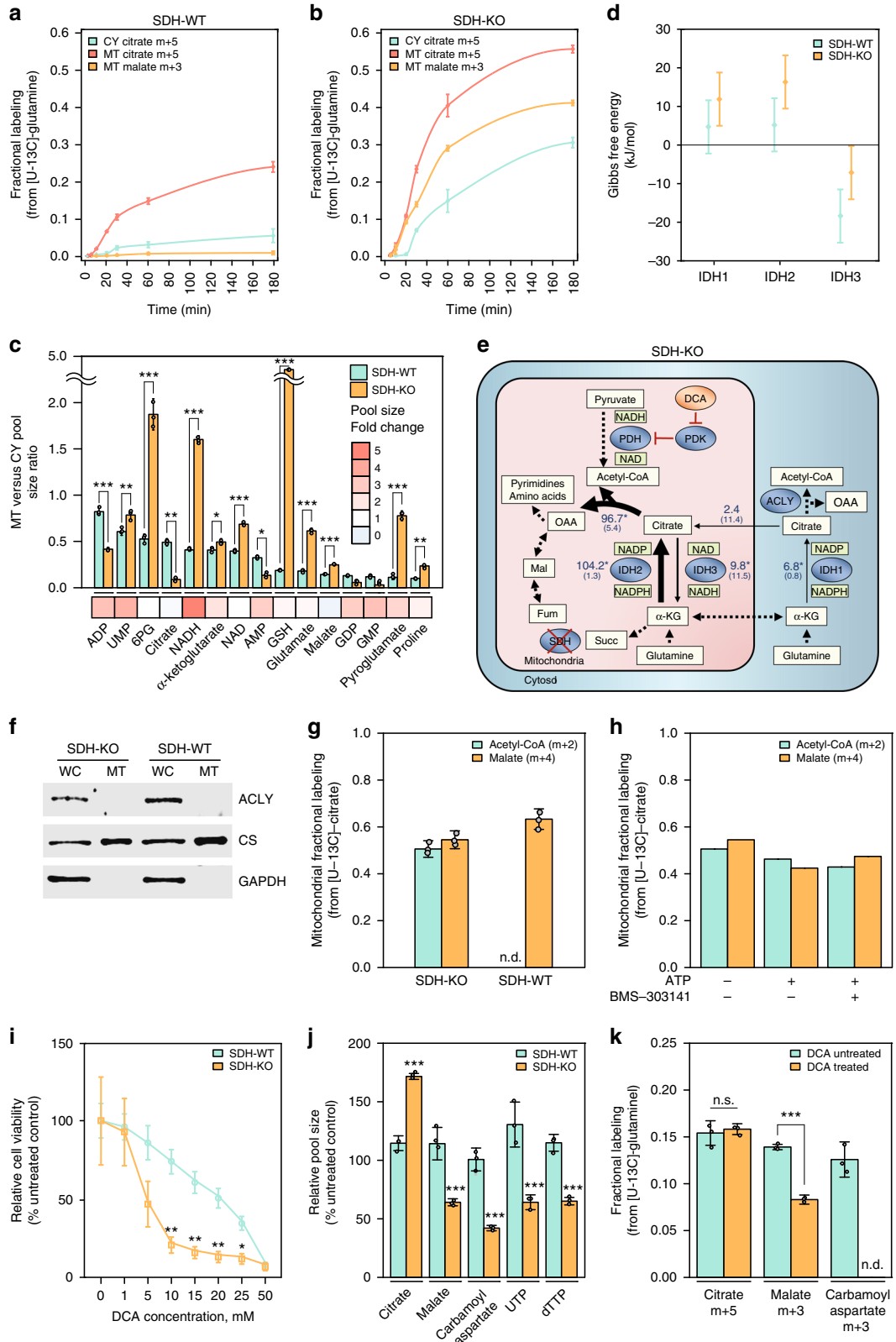

pyrimidine biosynthesis only in the SDH-deficient cells, as indicated by the significant drop in the concentration of malate, carbamoyl aspartate (an intermediate in pyrimidine biosynthesis), UTP, and dTTP only in these cells ($t$-test $p$-values $< 0.001$; Fig. 5j). Consistently, feeding isotopic glutamine, we detected a significant drop in the m + 3 form of malate ($t$-test $p$-value $< 0.001$) and

carbamoyl aspartate (to a non-detectable level; and similarly to non-detectable levels of pyrimidines with the m + 3 labeling form) upon DCA treatment in the SDH-deficient cells (while there was no change in the m + 5 form of citrate upon DCA treatment) (Fig. 5k). Taken together, our results support the suggested role of reverse CS flux in the survival and growth of SDH-defective cells.

**Fig. 5** Reversed CS flux supports cell survival and growth in SDH-deficient cells. **a, b** Isotopic labeling kinetics of cytosolic and mitochondrial citrate m + 5 and mitochondrial malate m + 3 when feeding SDH-WT (**a**) and SDH-KO cells (**b**) with [U-13C]-glutamine. **c** Mitochondrial to cytosolic metabolite pool size ratios (y-axis) in SDH-WT (green) and SDH-KO cells (orange); average fold change in whole-cell pool size upon SDH deficiency is indicated by color (x-axis). **d** Gibbs free energy of mitochondrial (IDH2/3) and cytosolic (IDH1) IDH isozymes (in the oxidative direction) in SDH-proficient (green) and deficient cells (orange). **e** Mitochondrial and cytosolic fluxes in SDH-KO cells, showing percentages from citrate synthase flux in the SDH-positive cells. Arrow represents the direction of net flux; number represents net flux in the direction of the arrow and number in parenthesis correspond to the backward flux. Confidence intervals for estimated fluxes shown in Supplementary Data 8. **f** Protein expression of mitochondrial and cytosolic markers in isolated mitochondria from SDH-WT and KO cells. **g** Fractional labeling of acetyl-CoA (m + 2) and malate (m + 4) in isolated mitochondria cultured with [U-13C]-citrate. **h** Fractional labeling of acetyl-CoA (m + 2) and malate (m + 4) in isolated SDH-KO mitochondria cultured with [U-13C]-citrate with or without ATP and ACLY inhibitor (BMS-303141). **i** Relative cell viability for SDH-WT (green) and SDH-KO cells (orange) following the addition of increasing concentration of dichloroacetate (DCA). XTT cell viability assay was performed 72 h after DCA treatment. **j** Relative metabolite pool sizes in SDH-WT (green) and SDH-KO cells (orange) with 10 mM DCA (24 h); compared to untreated control. **k** Metabolite fractional labeling from [U-13C]-glutamine in SDH-KO cells without (green) or with (orange) 10 mM DCA (24 h). *$P < 0.05$, **$P < 0.01$, and ***$P < 0.001$ by multiple t-test analysis with FDR correction (**f**) or by two-sample t-test (**c**, **e**, **g**, and **h**). n.d. not detected. Data are mean ± SD, $n = 3$ independent biological replicates

## Discussion

Fractionation of cells in a matter of seconds before quenching of metabolism is essential to minimally perturb metabolism and reliably quantify metabolite pools and isotopic labeling. The employed fractionation approach trades the purity of the derived subcellular fractions for speed: It takes ~25 s to complete and hence enables reliable measurement of mitochondrial and cytosolic metabolite pools with a fast turnover rate, though provides subcellular fractions that are not entirely pure (with ~10% cross-contamination between the mitochondrial and cytosolic fractions). The cross-contamination between the mitochondrial and cytosolic fractions was resolved using computational deconvolution. While nuclei remain intact within the mitochondrial fraction, it is not expected to bias the metabolomics measurements performed in the mitochondrial fraction due to the free diffusion of small molecules through NPCs into the cytosolic fraction (hence inferred cytosolic pool sizes represent the total abundance of metabolites in cytosol and nucleus, effectively considered as a single pool)[23,24]. A potential bias in the mitochondrial and cytosolic measurements is due to metabolite pools from other organelles such as endoplasmic reticulum, Golgi apparatus, peroxisomes, and lysosomes. Overall, the employed fractionation method is a complementary approach to existing methods for separating pure mitochondria as a basis for metabolomics studies via a substantially longer and potentially perturbative process[20].

Here, we presented a spatial-fluxomics approach for inferring metabolic fluxes in mitochondria and cytosol by combining isotope tracing in intact cells with rapid cell fractionation. Analyzing metabolite labeling before convergence to isotopic steady state enables direct observation of metabolite production in mitochondria and cytosol (based on the gradual accumulation of labeled metabolites in the mitochondrial or cytosolic pool) and metabolite transport across the mitochondrial membrane (e.g., see Fig. 2c–f). We apply a flux estimation method that is conceptually similar to non-stationary Metabolic Flux Analysis (MFA)[40,41] though utilizing measurements of metabolite isotopic labeling specifically in mitochondria and cytosol rather than whole-cell isotopic labeling measurements. Thermodynamic analysis of reaction Gibbs free energy in mitochondria and cytosol (computed based on derived compartment-specific metabolite pool sizes) is integrated with the flux modeling approach to constrain flux directionality and forward/backward flux ratios. This builds upon previous work on thermodynamic modeling utilizing whole-cell metabolite concentration measurements[42–50]. The combined flux and thermodynamic analysis provide unique means to observe flux through cytosolic and mitochondrial isozymes and between isozymes localized in the same compartment (IDH1-3).

Applying the spatial-fluxomics approach to re-examine reductive glutamine metabolism in HeLa cells, we find a previously unappreciated role of IDH1 as the sole net contributor of carbons to fatty acid biosynthesis under standard normoxic conditions in these cells. Consistent with previous reports, we find that the relative contribution of whole-cell level reductive IDH to producing citrate increases in hypoxia[21] and in cells with defective mitochondria[22]. However, our analysis shows that this is due to a drop in glycolytic flux into the TCA cycle producing citrate in hypoxia, rather than due to an actual increase in reductive IDH flux. In fact, we show that the mitochondrial reductive IDH2 flux drops more than 2-fold in hypoxia (leading to an increase in the relative contribution of IDH1 to the total cellular αKG reductive flux).

Utilizing spatial-fluxomics, we identified a new route through which SDH-defective cells support pyrimidine biosynthesis—reductive glutamine metabolism via mitochondrial IDH2, followed by a reverse flux through CS to produce OAA, which is further utilized for biosynthetic purposes. While net CS flux in the reverse direction toward OAA is thermodynamically unfavorable under standard biochemical conditions ($\Delta G^{o'}$ of $\sim -35$ kJ mol$^{-1}$), it was recently found to operate in this direction in some anaerobic bacteria, where the ratio of CoA/acetyl-CoA ratio was especially high[51]. Here, the concentration of the mitochondrial OAA is expected to markedly drop in SDH-deficient cells due to a blockage in the oxidative TCA cycle; and the level of mitochondrial acetyl-CoA is expected to decrease due to the pseudo-hypoxic state of SDH-deficient cells that block glycolytic flux into the TCA cycle[52]. Unfortunately, the concentration of OAA could not be readily quantified by LC-MS analysis in the mitochondrial fraction, preventing the direct thermodynamic assessment of the direction of net flux through CS. Therefore, the reverse CS flux in SDH-deficient cells was confirmed by a series of experiments done on a whole-cell level, isolated mitochondria, and cell lysates, while utilizing a chemical inhibitor of ACLY or genetically silencing its expression. Notably, our analysis cannot preclude the presence of an alternative mitochondrial pathway converting reductively synthesized citrate to OAA. For example, reductively synthesized citrate in mitochondria can produce pyruvate (via the production of itaconyl-CoA/citramalyl-CoA[53]; Supplementary Fig. 13), which in turn can produce OAA through PC. However, the isotopic glutamine tracing experiment rules out a potential activation of this pathway in the SDH-deficient cells (which should result in the production of malate m + 2 rather than the observed m + 3 form; Supplementary Fig. 13).

While we find that reverse CS flux supports OAA production and hence amino acid and pyrimidine biosynthesis, the fate of generated mitochondrial acetyl-CoA remains unclear. Mitochondrial acetyl-CoA can be converted to acetate and exported out to cytosol[54], acetylate mitochondrial proteins[55], participate in amino acid metabolism[56], or be employed to synthesize ketone bodies (acetoacetate, acetone, and β-hydroxybutyrate)[57].

Consistent with our previous observation, we found increased labeling of acetylated amino acids upon SDH deficiency (Supplementary Fig. 14a). Interestingly, we found that SDH-KO mitochondria have a ~2.3-fold larger acetoacetate pool and a ~1.7-fold increased acetoacetate m + 4 (when fed with [U-$^{13}$C]-glutamine) compared to that of the wild-type (Supplementary Fig. 14b-c). This implies that ketogenesis, triggered by citrate cleavage in SDH-KO mitochondria, can contribute to tumor growth and survival. For example, acetoacetate can be converted to β-hydroxybutyrate via BDH1 that exerts an anti-inflammatory response[58]. Exploring whether the induced acetylation and production of acetoacetate play an important role in the viability of SDH-KO cells may reveal new drug targets for SDH-deficient cancer therapy.

Taken together, we demonstrated that the spatial-fluxomics approach provides novel insights into compartmentalized fluxes that could be used to identify specific vulnerabilities in tumors. We expect this spatial-fluxomics approach to be a highly useful tool for studying the metabolic interplay between mitochondria and cytosol and facilitate an in-depth understanding of metabolic rewiring in cancer.

## Methods

**Materials**. Antibodies to citrate synthetase (ab96600), GAPDH (ab8245), IDH1 (ab94571), and IDH2 (ab55271) were from Abcam. Anti-Tubulin Antibody (MAB1637) was from EMD Millipore. IRDye 680RD goat anti-rabbit (926–68071) and IRDye 800CW goat anti-mouse (926–32210) secondary antibodies were from LI-COR. Primary and secondary antibodies were used at 1:1000 and 1:15,000 dilutions, respectively. Digitonin (D141), α-ketoglutaric acid (K3752), citric acid (C7129), malic acid (M1000), and sodium dichloroacetate (347795) were purchased from Sigma; glutamic acid (41–217–25)—from Biological Industries. [U-$^{13}$C]-glutamine (CLM-1822-H-0.1), [1-$^{13}$C]-glutamine (CLM-3612), and [U-$^{13}$C]-glucose (CLM-1396-1) were purchased from Cambridge Isotope Laboratories.

**Cell culture**. HeLa cells were obtained from ATCC while SDH-WT (*Sdhb$^{fl/fl}$*) and homogenous SDH-KO (*Sdhb$^{Δ/Δ}$*) cells were the gifts of Prof. Eyal Gottlieb (Technion). *Sdhb$^{fl/fl}$* are immortalized primary kidney epithelial cells from genetically modified mice containing loxP sites flanking exon 3 of the endogenous *Sdhb* gene. *Sdhb$^{Δ/Δ}$* was previously generated by infection of *Sdhb$^{fl/fl}$* cells with recombinant adenovirus expressing Cre recombinase. None of the cell lines used here are listed in the database of commonly misidentified cell lines maintained by ICLAC and NCBI Biosample. Both cell lines were confirmed to be mycoplasma free by EZ-PCR Mycoplasma Test Kit (Biological Industries, 20-700-20). Cell lines were grown in Dulbecco's modified eagle medium (DMEM) without pyruvate (Biological Industries) supplemented with 10% (v/v) heat-inactivated dialyzed fetal bovine serum (Sigma), 4.5 g l$^{-1}$ D-glucose, 3 mM L-glutamine, 100 U ml$^{-1}$ penicillin, and 100 μg ml$^{-1}$ streptomycin in a 5% CO$_2$ incubator at 37 °C. For hypoxic culture, HeLa cells and upper mentioned DMEM media were kept in Whitley H35 HypOxystation (don Whitley Scientific) at 37 °C, 1% O$_2$, 5% CO$_2$, and 94% N$_2$ for 24 h before further experiments. Cells were evenly seeded at 80,000 cells per 6-cm tissue culture dish and allowed to stabilize for 48 h. For all experiments, medium was replaced 6 h before subcellular fractionation and/or isotope tracer addition.

**Rapid subcellular fractionation method**. Cells grown in 6-cm tissue culture dishes were harvested at 80% confluency. After removing culture medium from the dish, cells were washed twice with 2 ml of ice-cold phosphate buffer saline (PBS) pH 7.4, scraped from culture dishes using a cell lifter, and collected into a microcentrifuge tube in 1 ml of ice-cold PBS. After brief centrifugation (13,500 g, 4 °C, 10 s), supernatant was removed and cell pellet was resuspended in 1 ml of ice-cold 1 mg ml$^{-1}$ digitonin in PBS and triturated 5 times using a P1000 micropipette. After second centrifugation (13,500 g, 4 °C, 10 s), supernatant and pallet were collected as "cytosolic fraction" and "mitochondrial fraction", respectively. Metabolism was quenched and metabolites were extracted by immediately adding 4 mL of −80 °C 62.5:37.5 (v/v) methanol:acetonitrile (cytosolic fraction) or 100 μL of −80 °C 50:30:20 (v/v/v) methanol:acetonitrile:water (mitochondrial fraction). "Whole-cell sample" was collected after the first centrifugation by adding 1 mL of −80 °C 50:30:20 (v/v/v) methanol:acetonitrile:water. Metabolite samples were stored at −80 °C until analysis.

**Confocal microscopy**. For immunofluorescence, HeLa cells were incubated with 500 nM MitoTracker Deep Red FM for 30 min at 37 °C. After incubation, cells were fractionated as described above. Each pellet from the first and second centrifugation (whole-cell sample and mitochondrial fraction) was fixed with 4% paraformaldehyde for 1 h, blocked with 2% BSA/0.5% Triton in PBS for 2 h at room temperature.

The cells were then incubated with GAPDH antibody (1/100) in blocking solution overnight at 4 °C, followed by further incubation at room temperature for 1 h with an anti-mouse AlexaFluor® 488 (1/250). Nuclear DNA was labeled in blue with an antifade mounting medium with DAPI (Vectashield). Confocal microscopy was performed with an LSM 710 laser scanning microscope (Zeiss).

**Immunoblotting**. For quantitative western blot assay, each pellet from the first and second centrifugation (whole-cell sample and mitochondrial fraction) was resuspended in 1 mL of ice-cold PBS, sonicated three times for 10 s on ice, and mixed with 250 μL of 5x Laemmli sample buffer. Supernatant from the second centrifugation (cytosolic fraction) was sonicated and mixed with 250 μL of 5x Laemmli sample buffer. Samples were heated at 95 °C for 5 min and centrifuged (16,000 g, 1 min) before loading on the gel. Standard curve was generated by loading different volume of the whole-cell sample on the gel (0.8, 5, 10, and 30 μL for CS; 0.8, 5, 20, and 30 μL for GAPDH). For accurate quantification of recovery and contamination rate of each fraction, quantitative infrared western blot assay was performed in triplicate using fluorescent secondary antibody and Odyssey Fc Imaging System (LI-COR Biosciences). All uncropped images from western blot can be found in Supplementary Fig. 15.

**Generation of HeLa cells with IDH1/2 knockdown**. HeLa cells were transfected using linear 25-kDa polyethylenimine (PEI; Polysciences)[59]. pLKO.1 shRNA vectors targeting IDH1 had sequence of CCGGGCTGCTTGCATTAAAGGTTTAC TCGAGTAAACCTTTAATGCAAGCAGCTTTTT (IDH1a and IDH1b; TRCN0000027298), and IDH2 had shRNA sequence CCGGCCAAGAACACCAT ACTGAAAGCTCGAGCTTTCAGTATGGTGTTCTTGGTTTTTG (IDH2a; TRCN0000229778) or CCGGTGATGAGATGACCCGTATTATCTCGAGATAAT ACGGGTCATCTCATCATTTTTG (IDH2b; TRCN0000229434). For control, pLKO.1 scrambled control vector (Addgene) was used. Individual clones were selected at least five passages after puromycin (2 μg ml^−1) treatment.

**siRNA transfections**. SDH-KO cells were transfected with Lipofectamine RNAi-MAX (Invitrogen) and siRNA pool targeting murine *ACLY* (Dharmacon #L-040092–01) or a non-targeting control (Dharmacon #D-001810-01-20) at a final concentration of 20 nM[60].

**Cell cytotoxicity assay**. An XTT-based cytotoxicity assay was performed as recommended by the manufacturer (Biological Industries, 20-300-1000). SDH-proficient and deficient cells were plated in a 96-well dish and incubated for 72 h with or without DCA (0, 1, 5, 10, 20, 25, and 50 mM). The XTT reagent was mixed with phenazine methosulfate immediately before labeling cells and incubated for 2 h at 37 °C and 5% CO$_2$. The 450 nm absorbance was measured using Epoch Microplate Spectrophotometer (BioTek Instruments, Inc.). Relative cell viability was calculated according to the following equation:

$$\text{Relative cell viability} (\%) = \left( \frac{\text{cells with DCA} - \text{blank}}{\text{cells without DCA} - \text{blank}} \right) \times 100$$

**Cell extract and enzyme assay**. Cells were resuspended in 20 mM Tris-HCl (pH 8), 5 mM DTE[51], lysed by sonication on ice for 45 s (15 s burst 10 s interval, 2 kJ total energy input, 20 kHz, 20% amplitude). Cell debris and insoluble material were removed by centrifugation (16,000 × g, 20 min, 4 °C). Resulting cell lysate was incubated at 37 °C with 200 r.p.m. agitation for 10 minutes within an assay mixture containing 50 mM Tris-HCl (pH 8), 2.5 mM DTE, 5 mM MgCl2, 2 mM CoA, 2 mM NADH, and 2 mM [U-$^{13}$C]-citrate. CoA-and NADH-dependent formation of acetyl-CoA (m + 2) and malate (m + 4) were measured via LC-MS.

**Mitochondria isolation and isotope tracing**. Mitochondria were prepared with the magnetic beads method (Mitochondria Isolation Kit; Miltenyi Biotec,), and the mitochondrial pellets were reconstituted in assay buffer (125 mM KCl, 10 mM Tris/MOPS, 0.1 mM EGTA/Tris, 1 mM Pi, pH 7.4) supplied with indicated nutrients and tracer[3]. For citrate tracing, 40 μM [U-$^{13}$C]-citrate, 40 μM NADH, and 40 μM CoA with or without 40 μM ATP and 50 μM ACLY inhibitor were added to the assay buffer. Mitochondria were incubated in the tracing buffer for 10 min, at 37 °C with 200 r.p.m. agitation in a heat block.

**LC-MS analysis**. Chromatographic separation was achieved on a SeQuant ZIC-pHILIC column (2.1 × 150 mm, 5 μm, EMD Millipore). Flow rate was set to 0.2 ml min$^{-1}$, column compartment was set to 30 °C, and autosampler tray maintained 4 °C. Mobile phase A consisted of 20 mM ammonium carbonate and 0.01% (v/v) ammonium hydroxide. Mobile Phase B was 100% acetonitrile. The mobile phase linear gradient (%B) was as follows: 0 min 80%, 15 min 20%, 15.1 min 80%, 23 min 80%. A mobile phase was introduced to Thermo Q Exactive mass spectrometer with an electrospray ionization source working in polarity switching mode. Ionization source parameters were following: sheath gas 40, auxiliary gas 10, spray voltage −3.25 kV or +4.25 kV, capillary temperature 325 °C, S-lens RF level 50, auxiliary gas temperature 50 °C. Metabolites were analyzed in the range 70–1000 m/z. Positions of metabolites in the chromatogram were identified by corresponding pure chemical standards. Data were analyzed with MAVEN[61]. Absolute

metabolite pool sizes in whole-cell extracts were quantified by using isotope-ratio with chemical standards[25]. Metabolite pool sizes are expressed per number of cells and total cell volume, measured via a Multisizer Coulter Counter (Beckman Coulter).

For the analysis of fraction purity, HeLa cells were incubated with 50 nM TMRM for 4 h in DMEM at 37 °C, followed by a further incubation in TMRM-free medium for 4 h to remove unbound dye. After incubation, cells were fractionated as described above. TMRM and glucose-6-phosphate abundance in each fraction was normalized to the corresponding value in whole-cell samples.

**Metabolite pool sizes in mitochondria and cytosol**. Relative metabolite pool sizes in the derived mitochondrial and cytosolic fractions out of the total whole-cell pool were quantified via isotope-ratio. Specifically, isotopically labeled metabolite extracts were obtained by feeding HeLa cells with [U-13C]-glucose and [U-13C]-glutamine for 24 h, deriving whole-cell metabolite extracts as described above. The labeled metabolite extracts were used as internal standards for the LC-MS analysis of metabolite extracts from the mitochondrial and cytosolic fractions. For each metabolite, we denote the ratio of its labeled-to-unlabeled form determined by LC-MS in the internal standard extracts by $R1$, and the ratio of its labeled-to-unlabeled form in an analyzed subcellular fraction (either mitochondrial or cytosolic) mixed with the internal standards by $R2$. The ratio of the metabolite pool size in that subcellular fraction is calculated based on $(1 - R2/R1)/(R2 + R2/R1)$. Considering that the ratio of mitochondrial versus cytosol pool size varies significantly between metabolites, we applied multiple dilution ratios between the internal standards and mitochondrial/cytosolic fractions (1:1, 1:5, and 1:10). We denote the inferred relative pool sizes in the mitochondrial and cytosolic fractions out of the total whole-cell pool by $P_m$ and $P_c$, respectively.

**Isotope tracing coupled with rapid fractionation**. Isotope tracing was performed by feeding exponentially growing HeLa cells with either [U-13C]-glucose or [U-13C]-glutamine. To minimize the perturbation to cells by the replacement of the media (which is especially important when tracking transient labeling kinetics), we replaced just half of the media volume with DMEM containing the isotopic substrate. Cells were fed with labeled substrates for different time (2 min, 5 min, 10 min, 20 min, 30 min, 1 h, 3 h, 12 h, and 24 h), rapidly fractionated, and metabolism was quenched immediately (as described above). LC-MS was used to measure the mass-isotopomer distribution of metabolites in each derived subcellular fraction. Measured mass-isotopomer distributions were corrected for natural abundance of 13C[62].

**Deconvolution of metabolite pool sizes and isotopic labeling**. We denote the relative abundance of the cytosolic marker detected in the mitochondrial fraction out of that detected in the cytosolic fraction by $\alpha$, and the relative abundance of the mitochondrial marker detected in the cytosolic fraction out of that detected in the mitochondrial fraction by $\beta$. Based on the analysis of fraction purity via small-molecule markers (Fig. 1c, d), $\alpha$ is set to 0.13 and $\beta$ is set to 0.11, with a standard deviation of 0.03. The deconvolution of measured metabolite pool sizes in the mitochondrial and cytosolic fraction is based on the following linear equations:

$$P_c = 1 \cdot X_c + \beta \cdot X_m$$
$$P_m = \alpha \cdot X_c + 1 \cdot X_m$$

where $X_m$ and $X_c$ denote the ratio of metabolite pool size per unit of the mitochondrial and cytosolic marker signals, respectively. The deconvoluted metabolite pool size in the cytosol is the sum of the cytosolic pool size in the cytosolic fraction ($X_c$) and the cytosolic pool size contaminating the mitochondrial fraction ($\alpha X_c$) and is denoted by $P'_c$. Similarly, the deconvoluted pool size in mitochondria is denoted by $P'_m$. Solving these equations we get: $P'_c = \frac{P_c - \beta P_m}{1 - \alpha\beta}(1 + \alpha)$, $P'_m = \frac{P_m - \alpha P_c}{1 - \alpha\beta}(1 + \beta)$. To account for noise in the experimental measurements, we generated a distribution of possible mitochondrial and cytosolic pool sizes ($P_m$ and $P_c$) and protein marker signals ($\alpha$ and $\beta$), based on the assumption of Gaussian noise (considering the empirical standard deviation in the experimental measurements), and calculated the deconvoluted pool sizes based on the mean and standard deviation of this distribution. For metabolites whose whole-cell absolute pool sizes were measured, the absolute pool size in each compartment (per number of cells) was determined by multiplying the deconvoluted relative pool sizes in cytosol ($P'_c$) and in mitochondria ($P'_m$) by the whole-cell relative pool size.

The deconvolution of mass-isotopomer distributions measured in the mitochondrial and cytosolic fractions is performed via a similar approach, though utilizing prior measurement of the relative metabolite pool sizes in each fraction ($P_m$ and $P_c$, respectively). Specifically, we denote the mass-isotopomer distributions for a metabolite measured by LC-MS in the mitochondrial and cytosolic fractions by the vectors $\overline{I_m}$ and $\overline{I_c}$, respectively (denoting the fraction of the metabolite pool having zero, one, two, etc. labeled carbons). The relative pool size of each mass-isotopomer in the mitochondrial and cytosolic fraction (out of the whole-cell pool of that mass-isotopomer) is calculated by multiplying these measured mass-isotopomers distributions by the relative metabolite pool sizes in each compartment. Deconvolution of the relative pool size of each mass-isotopomer in the mitochondrial ($\overline{I_m} \cdot P_m$) and cytosolic fraction ($\overline{I_c} \cdot P_c$) is performed as

described above, where the deconvoluted mass-isotopomer pool sizes in mitochondria and cytosol denoted $\overline{I'_m}$ and $\overline{I'_c}$.

**Thermodynamic analysis of IDH isozymes**. Reaction Gibbs free energy was calculated based on $\Delta_r G' = \Delta_r G'^o + RT \cdot ln(Q)$, where $\Delta_r G'^o$ is the Gibbs free energy at standard conditions, $Q$ is the reaction quotient, $R$ being the gas constant, and $T$ the temperature in Kelvin. The standard Gibbs free energy for NADH and NADPH-dependent IDH was inferred via Component Contribution[63]. Mitochondrial NADH:NAD+ ratios were measured utilizing chemical standards for NADH and NAD + [64], finding a ratio of 0.015 in HeLa cells grown under normoxia, in accordance with previous reports[20,65] (Supplementary Data 2). Due to a difficulty to reliably measure compartmentalized NADP + pool sizes, we utilized previously reported mitochondrial NADPH:NADP+ ratio of ~100 (with a coefficient of variance of 30% to account for uncertainty in the actual value)[18,20,65], and cytosolic NADPH:NADP+ ratio between 30 and 100[66,67]. A $CO_2$ concentration of 1.2 mM was assumed based on Henry's law and in accordance with literature. The analysis of Gibbs free energy accounts for the noise in the estimation of standard Gibbs free energy and uncertainty regarding exact metabolite pool sizes and redox factors in each compartment. Specifically, we generated a distribution of possible standard Gibbs free energies for the NADH and NADPH-dependent IDH reactions and reactant pool sizes based on the assumption of Gaussian noise (considering the empirical standard deviation in the experimental measurements) and calculated the Gibbs free energy based on the mean and standard deviation of this distribution. Notably, the analysis does not depend on an assumption regarding the specific volume of mitochondria or cytosol (only on the ratio between product to substrate pool sizes). The flux-force relationship entails that the ratio between forward flux is proportional to Gibbs free energy[29]: $\Delta_r G' = -RT ln(J^+/J^-)$, where $J^+$ being the forward flux and $J^-$ the backward flux.

**Compartmentalized metabolic flux analysis (MFA)**. To infer cytosolic versus mitochondrial metabolic fluxes involved in citrate metabolism, we constructed a compartmentalized flux model consisting of distinct metabolite pools and isozymes in each compartment (Supplementary Fig. 4). Kinetic Flux Profiling (KFP)[28] was applied to identify a flux distribution that optimally fits the following experimental datasets: (i) Isotopic labeling kinetics of citrate, malate, and αKG in mitochondria and cytosol when feeding cells with [U-13C]-glutamine (inferred via the deconvolution method described above; $\overline{I'_m}$ and $\overline{I'_c}$; Supplementary Data 3–7); (ii) measured pool size of citrate in mitochondria and cytosol (Supplementary Data 2); (iii) ratio of forward-to-backward fluxes through the various IDH isozymes in mitochondria and cytosol, inferred based on thermodynamics (see above).

Non-convex optimization was used to identify a vector of fluxes $\mathbf{v}$ that maximizes the log-likelihood of measured mass-isotopomer labeling kinetics of citrate in mitochondria and cytosol. We denote by by $X^M_{cit,j}(t)$ and $X^C_{cit,j}(t)$ the relative abundance of the $j^{th}$ mass-isotopomer of citrate (i.e. citrate having $j$ labeled carbons) in mitochondria and cytosol, respectively, after $t$ minutes feeding with isotopic glutamine. The expected mass-isotopomer distribution of citrate in mitochondria and cytosol after $t$ minutes feeding with isotopic glutamine is denoted $Y^M_{cit,j}(t,v,)$ and $Y^C_{cit,j}(t,v,)$. Maximum likelihood estimate of fluxes are obtained by minimizing the variance-weighted sum of squared residuals between measured and computed mass-isotopomer distributions, where $\sigma^M_j(t)$ and $\sigma^C_j(t)$ represent the standard deviation in the measurement of the relative abundance of the $j^{th}$ mass-isotopomer of citrate in in mitochondria and cytosol, respectively, $t$ minutes after feeding isotopic glutamine:

$$\min_v \sum_{t \in T} \sum_{j \in \{4,5\}} \left( \frac{X^M_{cit,j}(t) - Y^M_{cit,j}(t,v)}{\sigma^M_j(t)} \right)^2 + \left( \frac{X^C_{cit,j}(t) - Y^C_{cit,j}(t,v)}{\sigma^C_j(t)} \right)^2$$

s.t.

$$v_{CS} + v_{CIT\_B} + v_{IDH2\_R} + v_{IDH3\_R} = v_{CIT\_F} + v_{IDH2\_O} + v_{IDH3\_O} \quad (1)$$

$$v_{CIT\_F} + v_{IDH1\_R} = v_{CIT\_B} + v_{IDH1,O} + v_{AcCoa} \quad (2)$$

$$v \geq 0 \quad (3)$$

$$(b_{IDHk} - 2\sigma_{IDHk}) \cdot v_{IDHk\_O} \leq v_{IDHk,R} \leq (b_{IDHk} + 2\sigma_{IDHk}) \cdot v_{IDHk\_O}, k = \{1,2,3\} \quad (4)$$

where equation (Eq. 1) enforce stoichiometric mass-balance for mitochondrial citrate and equation (Eq. 2) for cytosolic citrate. $v_{CS}$ denotes the rate of citrate synthase; $v_{CIT\_F}$ and $v_{CIT\_B}$ denote the rate of citrate transport from mitochondria to cytosol and back, respectively; $v_{IDHk\_O}$ and $v_{IDHk\_R}$ denote the rate of the $k^{th}$ isozyme of IDH (i.e. IDH1, IDH2, or IDH3) in the oxidative and reductive direction, respectively; $v_{AcCoa}$ denotes the rate of cellular demand for cytosolic acetyl-CoA. All reactions in the model are defined as irreversible, having non-negative flux (Eq. 3). Backward-to-forward flux ratio is constrained in Eq. 4, where $b_{IDHk}$ and $\sigma_{IDHk}$ represent the backward-to-forward flux ratio and standard deviation for the $i^{th}$ isozyme of IDH, respectively, inferred based on thermodynamic analysis (see above). The isotopic labeling kinetics of

mitochondrial and cytosolic metabolites ($Y^M_{cit,j}(t, \nu,)$ and $Y^C_{cit,j}(t, \nu,)$) were simulated given a flux vector $\nu$ via a set of ordinary differential equations[41,68]:

$$\frac{dY^M_{cit,4}(t,\nu)}{dt} = \frac{1}{u^M_{cit}}\left(\nu_{CS}X^M_{mal,4}(t) + \nu_{CIT\_B}Y^C_{cit,4}(t,\nu) - (\nu_{CS} + \nu_{IDH2\_R} + \nu_{IDH3\_R} + \nu_{CIT\_B})Y^M_{cit,4}(t,\nu)\right)$$

(5)

$$\frac{dY^M_{cit,5}(t,\nu)}{dt} = \frac{1}{u^M_{cit}}\left((\nu_{IDH2\_R} + \nu_{IDH3\_R})X^M_{aKG,5}(t) + \nu_{CIT\_B}Y^C_{cit,5}(t,\nu) - (\nu_{CS} + \nu_{IDH2\_R} + \nu_{IDH3\_R} + \nu_{CIT\_B})Y^M_{cit,5}(t,\nu)\right)$$

(6)

$$\frac{dY^C_{cit,4}(t,\nu)}{dt} = \frac{1}{u^C_{cit}}\left(\nu_{CIT\_F}Y^M_{cit,4}(t,\nu) - (\nu_{CIT\_F} + \nu_{IDH1\_R})Y^C_{cit,4}(t,\nu)\right)$$

(7)

$$\frac{dY^C_{cit,5}(t,\nu)}{dt} = \frac{1}{u^C_{cit}}\left(\nu_{CIT\_F}Y^M_{cit,5}(t,\nu) + \nu_{IDH1\_R}X^C_{aKG,5}(t) - (\nu_{CIT\_F} + \nu_{IDH1\_R})Y^C_{cit,5}(t,\nu)\right)$$

(8)

where the momentary change in the fractional labeling of the m + 4 and m + 5 mass-isotopomers of mitochondrial and cytosolic citrate (left hand side in the above equations) computed based on the difference between the momentary production and consumption of the different mass-isotopomers of citrate (term in parenthesis on right hand side of the equations), normalized by the citrate pool size mitochondria ($u^M_{cit}$) and cytosol ($u^C_{cit}$). Modeling the labeling kinetics of mitochondrial citrate (Equations 5–6), the mitochondrial acetyl-CoA is assumed non-labeled (due to the contribution of malic enzymes to the production of mitochondrial pyruvate being substantially smaller than that of glycolysis). The latter is evident by the small fractional labeling of mitochondrial citrate m+6 in all experiments (<0.5%), produced via citrate synthase from malate m + 4 and acetyl-CoA m + 2 (Supplementary Data 6). Notably, in the SDH knockout cells, citrate m + 4 is not produced due to the defective activity of oxidative TCA cycle (Supplementary Data 6). Hence, analyzing metabolic fluxes in the SDH knockout cells, Equations 5–6 were changed to model the isotopic labeling kinetics of citrate m + 3 instead of citrate m + 4 (see Supplementary Fig. 8). Citrate m + 3 in these cells is produced in the mitochondria from malate m + 3 via citrate synthase.

The non-convex optimization problem was solved using Matlab's Sequential Quadratic Optimization (SQP), starting from multiple sets of random fluxes to overcome potential local minima. To compute confidence intervals for estimated fluxes, SQP was iteratively run to compute the maximum log-likelihood estimation while constraining the flux to increasing (and then decreasing) values (with a step size equal to 5% of the flux predicted in the initial maximum log-likelihood estimation)[68,69]. Confidence interval bounds were determined based on the 95% quantile of $\chi^2$-distribution with one degree of freedom.

**Reporting summary**. Further information on experimental design is available in the Nature Research Reporting Summary linked to this article.

## Code availability

All code used to conduct the research detailed in this manuscript is available on request from the corresponding author.

## Data availability

The authors declare that all the data supporting the findings of this study are available within the article and its Supplementary Information Files and from the corresponding author upon reasonable request. The source data underlying figures are provided as a Source Data file.

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

## Acknowledgements

We would like to thank Yoav Arava and Eyal Gottlieb for providing valuable comments on this manuscript. The research leading to these results has received funding from the European Research Council/ERC Grant Agreement No. 714738.

## Author contributions

W.D.L. and T.S. conceived and designed the study. W.D.L. and E.A. performed experiments. D.M. and W.D.L. performed LC-MS measurements. W.D.L., D.M. and T.S. analyzed the results. W.D.L. and T.S. wrote the manuscript. This project was supervised by T.S. All authors participated in editing and refining the analysis and the final version of this manuscript.

## Additional information

**Competing interests:** The authors declare no competing interests.

