## [Peer Review File · Nature Communications]

Reviewers' comments:

Reviewer #1 (Remarks to the Author):

The manuscript submitted by Lee et al. reports about a study of reductive glutamine metabolism in HeLa cells featuring a subcellular resolution. The authors applied a fractionation of mitochondria by digitonin permeabilization of the cytoplasmic membrane. Quenching and fractionation can be done within 25 seconds. Based on separate analyses of (conventional) whole cell fractions and mitochondria enriched fractions in combination with stable-isotope labeling and LC-MS, the authors could show that fatty acid biosynthesis through cytosolic reductive carboxylation from glutamine is the major producer even under normoxic conditions in HeLa cells. Based on kinetic flux profiling and due to quantitative metabolite measurements in mitochondria and cytosol, the authors could determine fluxes through IDH1,2 and 3. They also studied all fluxes under hypoxia and claim that the increase of cit M5 is due to reduced influx of glycolytic carbon into mitochondria. Finally, they studied metabolic fluxes in SDH deficient cells and claim that due to the break in the TCA cycle, reductive carboxylation produces citrate in mitochondria and is there converted to acetyl-CoA and oxaloacetate via reverse activity of citrate synthase(!). The manuscript is clearly and well written. Although the idea of determination of subcellular fluxes through selective permeabilization is not new and has been applied by others for mitochondrial flux analysis, the current study takes maximum advantage of this method and could determine quantitative metabolite amounts that can be applied for flux determination based on Gibbs free energy.

The most exciting element is obviously the reverse flux through citrate synthase (CS). And here I have my major doubts. Based on decades of text book knowledge this enzyme is known to be irreversible under physiological conditions. If CS can really operate in reverse, this would be a ground breaking discovery. A recent paper showing reversibility of CS in a thermophilic bacterium was published in Science (and cited by the authors). However, in this case the bacteria were cultivated at high temperature (>50°C) most probably providing energy for the reverse flux. Moreover, if Lee et al. really claim a reversibility of CS in mammalian cells, this needs to be shown in more detail and by a controlled enzymatic assay in vitro.

There are some more points that would need to be addressed before publication:

1. Reverse flux through CS has to be demonstrated in an in vitro assay.
2. The authors should take advantage of their assay and permeabilize cells and incubate the remaining functional mitochondria in the presence of a citrate tracer. They should then demonstrate that significant fractions of oxaloacetate and acetyl-CoA are produced inside mitochondria. They need to demonstrate that there is no residual ACLY activity left or use an ACLY inhibitor.
3. The authors should use their permeabilization assay and incubate functional mitochondria in the presence of a U13C glutamine tracer. If CS is reversible, they will observe accumulation of M3 malate in this setup. As in point 2 absence of ACLY activity needs to be ensured.
4. The authors should silence ACLY and show that M1 malate fractions are not effected when using a 1-13C1 glutamine tracer.
5. It is not clear how mitochondrial volume was determined. The volume is required for calculation of concentrations of mitochondrial metabolites and for Gibbs free energy determination. The volume of mitochondria can vary significantly between conditions and cell lines.
6. The title is misleading. While the authors exclusively studied HeLa cells, they generalize these results to cancer cells in common. They should reproduce their results in a few more well characterized cell lines such as A549 or HepG2 to test whether this observation is typical for the

studied cell model or common to cancer cells.

7. It is surprising that the authors found a high ratio of mitochondrial to cytosolic AMP (Fig 1h). I would expect the opposite under normal conditions. Please discuss.

8. There are no measurements for NADP levels. However, the ratio NADPH/NADP is critical for IDH1 and IDH2 activities. It is not clear why the NADH/NAD and NADPH/NADP ratios have been assumed and not measured. These ratios play a key role for some of the main results of the manuscript (e.g. reverse IDH1 flux under normoxia).

9. Maybe I missed, how acetyl-CoA was measured. The absolute amount of mitochondrial acetyl-CoA is important for the claimed reverse flux through CS.

10. Alanine seems to be low in mitochondria vs cytosol. However, previous publications show the opposite (e.g. Buescher et al 2015). Please discuss.

11. It is not clear how quantitative uptake and secretion fluxes have been measured.

Reviewer #2 (Remarks to the Author):

The authors present very interesting work about determination of compartmental fluxes at the cytosol/mitochondria interface using a new combination of powerful techniques that were partly developed for this purpose. A major step is a new method to determine compartmental levels of metabolites and their carbon labelling. Thereby, an important step is rapid subcellular fractionation followed by LC-MS analysis of metabolites. They focus on the metabolism of glutamine, a major nutrient in mammalian cell culture, in human HeLa and mouse cells exhibiting succinate dehydrogenase (SDH) deficient cells. The compartmental analysis of the level and labelling of many metabolites allows the application of an estimation method only relying on local data with a high resolution of a small part of metabolism of interest. This simplifies the estimation of fluxes considerably not requiring all kinds of detailed information of inter-compartment fluxes etc. Thus it looks more widely applicable with reasonable effort.

For the desired determination of real compartment concentrations the cellular and compartment volumes would have to be known. Concentrations are required for a thermodynamic analysis as carried out in this work. The authors took values from the literature but handled them with sufficient care, I think.

Using their dynamic labelling and compartmental metabolite analysis systems, they were able to elucidate the role of the three isocitrate dehydrogenase (IDH) isoenzymes that are partly localized in the mitochondria (IDH2 and IDH3) and in the cytosol (IDH1). IDH1 and IDH2 require NADP/NADPH as cosubstrates whereas IDH3 uses NAD/NADH. They could show that IDH1 provides acetyl-CoA for fatty acid biosynthesis via reduction of α -ketoglutarate to citrate. Under hypoxia the glycolytic flux to yield mitochondrial citrate was reduced by more than 90% compared to that under normoxia whereas the cytosolic reductive synthesis of acetyl-CoA remained constant. The flux of the citrate synthase was reverted in mouse cells lacking SDH leading to the production of mitochondrial acetyl-CoA and oxaloacetate.

The conclusions drawn are original and clearly justified on the basis of the data collected and their analysis.

Detailed metabolic findings of this type will aid in the identification of drug targets with related diseases, e.g. in cancer. I think, therefore, that the developed methods will be applied on a broader scale in the near future.

The work is generally carried out with great care and is presented clearly. There are, however, some questions and suggestions for improving clarity.

1. I think it should be clearly stated (also in the abstract) that a part of the analyses was made using human HeLa and another one using SDH-deficient mouse cells. Arguments supporting and justifying the comparison between the two species should be added.

2. The authors investigate carefully the errors introduced during sampling, quenching and metabolite analysis in the two compartments. Most of this work is summarized in Fig. 1. They show clearly that the mitochondria remain intact (compartment specific staining of TMRM and of citrate synthase). This does, however, not strictly prove that all other metabolites remain in the mitochondria to the same extent. They show that the fraction of metabolites where the levels of the sum of subcellular metabolites agree fully with whole cell levels is about 60% (Fig. 1e). About 90% of the metabolite levels agree to about 15% and better. Here it would be important to specify the positioning of the metabolites that are of central interest in this study, e.g. glutamine, citrate, malate, etc. Was a deconvolution also carried out for individual metabolites beyond those shown in Supplementary Table 2??
3. Similarly the dynamics of these individual metabolites (Fig. 1.f) would be of interest. How are the time constants related to the metabolic time constants (=flux/concentration) of the network part analyzed in detail? What impact does this have on the flux analysis?
4. The authors were obviously also looking at extracellular metabolites (see Fig. 3e and l. 167). Did the secretion and uptake of these or other metabolites have any influence on the dynamic labeling data? Ideally, show such data in an supplementary tables!
5. The potential fate of the OAA synthesized in the mitochondria of SDH-KO cells is discussed extensively. What could happen with the mitochondrial acetyl-CoA in this case, when the citrate synthase is running in the opposite direction?
6. Fig. 1h. A log x-scale would be more informative, particular for the metabolites in the upper part of the diagram.
7. L.271. this is actually a higher reductive flux of alpha-ketoglutarate that was estimated.
8. The SDH-WT and SDH-KO cells should clearly refer to the cells supplied by Dr. Gottlieb (l. 380).
9. Supplementary Fig. 1. I could not find a clear definition of the x-axis (standard). What was exactly used as standard?
10. Supplementary Table 1 needs a more detailed description (heading and units of data presented)
11. Typo, l.91 – molecule marker
12. Supplementary Fig. 2f. The CO₂ released during the synthesis of Uracil on the right hand side should be unlabeled contrary to the one on the left.

December 15, 2018

Dear reviewers,

We would like to thank you for the careful and thorough reading of this manuscript as well as the thoughtful comments and constructive suggestions, which have helped us improve the quality of the manuscript. The changes in the manuscript and point-by-point replies to reviewers' comments are summarized below.

Responses to reviewer 1's comments:

Comment 1: Reverse flux through CS has to be demonstrated in an in vitro assay.

We thank the reviewer for this suggestion. As suggested, we now demonstrate reverse citrate synthase (CS) activity *in vitro* with SDH-KO cell lysate. The main challenge in demonstrating citrate cleavage by CS in cell lysates is the presence of a cytosolic enzyme, ATP-citrate lyase (ACLY), catalyzing ATP-dependent citrate cleavage. To distinguish the activity of these two enzymes, we evaluated the enzymatic activities with/without adding ATP as recently performed by Mall et al.¹; and using an ACLY inhibitor (BMS-303141), as suggested by the reviewer in Comment #2-3.

Cell extracts were prepared as previously described¹ with some modifications. Briefly, cells were resuspended in 20 mM Tris-HCl (pH 8), 5 mM DTE, lysed by sonication on ice for 45 s (15 s burst 10 s interval, 2 kJ total energy input, 20 kHz, 20% amplitude). Cell debris and insoluble material were removed by centrifugation (16,000 × g, 20 min, 4°C). Resulting cell lysate was incubated at 37 °C with 200 r.p.m. agitation for 10 minutes within an assay mixture containing 50 mM Tris-HCl (pH 8), 2.5 mM DTE, 5 mM MgCl₂, 2 mM CoA, 2 mM NADH, and 2 mM [U-¹³C]-citrate. CoA- and NADH-dependent formation of acetyl-CoA (m+2) and malate (m+4) were measured via LC-MS (Methods).

We observed ~36% of acetyl-CoA and malate being labeled in their m+2 and m+4 form, respectively, indicating citrate cleavage flux in the SDH-KO lysate (see Fig. X1 below). To test the activity of reverse CS specifically, we added 1 mM of the ACLY inhibitor BMS-303141. Note that, the IC₅₀ of BMS-303141 is 8 μM; 20 μM was previously shown to completely inhibit ACLY activity², while here we utilized a 50-times higher concentration to assure complete inhibition. We found that, while the fractional labeling of acetyl-CoA and malate dropped upon ACLY inhibition, both acetyl-CoA m+2 and malate m+4 were produced – testifying for reverse CS flux. Notably, the abundance of acetyl-CoA m+2 and malate m+4, with or without the inhibitor, does not reflect the actual flux within the cells, which is affected by numerous factors, including the compartmentalized concentration of reactants and presence of metabolic regulators, etc. – whereas the main point here is to qualitatively demonstrate reverse CS activity. Expectedly, adding ATP (without the inhibitor) substantially increased the abundance of acetyl-CoA m+2 and malate m+4 due to

ACLY activity. Adding ATP while treating with the inhibitor did not increase the abundance of acetyl-CoA m+2 and malate m+4, further testifying that the high concentration of the inhibitor indeed provided complete inhibition of ACLY. Further evidence for citrate cleavage flux via reverse CS versus through ACLY is provided via isolated mitochondria and genetic silencing of ACLY (see below, and lines 293-294 in the Manuscript).

Fig. X1: Formation of acetyl-CoA (m+2) and malate (m+4) from [U-¹³C]-citrate cleavage catalyzed by cell extracts of SDH-deficient cells.

Comment 2: The authors should take advantage of their assay and permeabilize cells and incubate the remaining functional mitochondria in the presence of a citrate tracer. They should then demonstrate that significant fractions of oxaloacetate and acetyl-CoA are produced inside mitochondria. They need to demonstrate that there is no residual ACLY activity left or use an ACLY inhibitor.

As suggested, we isolated mitochondria from both SDH-WT and KO cells, incubated them with [U-¹³C]-citrate, and measured the fractional labeling of acetyl-CoA. Note that, due to the instability of oxaloacetate (OAA)³, we probe OAA labeling via malate (OAA is reduced to malate by malate dehydrogenase) as previously described^{1,4}. In addition, to remove cytosolic ACLY thoroughly, we took advantage of a magnetic bead-based mitochondrial isolation approach to retain highly pure and functional mitochondria (considering that while our fractionation method is more rapid and suitable for metabolomics experiments aimed to reveal physiological metabolite levels, the mitochondrial fraction consists of ~10% of cytosolic contamination).

Mitochondria were prepared with the magnetic beads method (Mitochondria Isolation Kit; Miltenyi Biotec), and the resulting mitochondrial pellets were reconstituted in assay buffer (125 mM KCl, 10 mM Tris/MOPS, 0.1 mM EGTA/Tris, 1 mM Pi, pH 7.4) supplied with indicated nutrients and tracer⁴. For citrate tracing, 40 μM [U-¹³C]-citrate, 40 μM NADH, and 40 μM CoA with or without 40 μM ATP and 50 μM ACLY inhibitor were added to the assay buffer. Mitochondria were incubated in the tracing buffer for 10 min, at 37 °C with 200 r.p.m. agitation in a heat block (Methods).

The purity and integrity of isolated mitochondria from SDH-WT and KO cells were verified via western blot, where CS but not ACLY (and GAPDH) was detectable in the isolated mitochondria (Fig. X2a). Reverse flux through CS in isolated mitochondria of SDH-KO cells is clearly evident by the m+2 labeled acetyl-CoA (~51%) and m+4 malate (~54%) (Fig. X2b). Notably, malate m+4 cannot be generated through oxidative metabolism of citrate since SDH activity is absent in these cells.

Expectedly, we do not detect evidence of reverse CS flux in SDH-WT mitochondria, as acetyl-CoA m+2 is non-detectable; SDH-WT mitochondria though do generate a substantial amount of malate m+4 (63%) through oxidative metabolism of citrate (Fig. X2b). The fact that acetyl-CoA m+2 is not produced in the SDH-WT mitochondria shows that no residual ACLY remains within the mitochondrial fraction. No ACLY remaining in mitochondria isolates from SDH-KO cells is further evident by the addition of ATP and ACLY inhibitor not affecting the fractional labeling of acetyl-CoA m+2 or malate m+4 (Fig. X2c). Overall, our results strongly confirm reverse CS flux in SDH-KO mitochondria (lines 280-293)

Fig. X2: SDH-deficient mitochondria catalyze citrate cleavage via reverse CS activity. (a) Protein expression of mitochondrial and cytosolic markers in isolated mitochondria from SDH-WT and KO cells. (b) Fractional labeling of acetyl-CoA (m+2) and malate (m+4) in isolated mitochondria cultured with [U-¹³C]-citrate. (c) Fractional labeling of acetyl-CoA (m+2) and malate (m+4) in isolated SDH-KO mitochondria cultured with [U-¹³C]-citrate with or without ATP and ACLY inhibitor (BMS-303141).

Comment 3: The authors should use their permeabilization assay and incubate functional mitochondria in the presence of a U13C glutamine tracer. If CS is reversible, they will observe accumulation of M3 malate in this setup. As in Comment 2, absence of ACLY activity needs to be ensured.

The metabolism of isolated mitochondria is not expected to represent metabolic activity under physiological conditions. The entire point of our spatial-fluxomics approach is to provide direct means to scrutinize compartment-specific flux under physiological conditions (without perturbing metabolic activities). Hence, while reverse CS activity in isolated mitochondria is observable (as shown in our reply to Comment #2), there is no *a priori* reason to assume that isolated mitochondria in these cells could activate the entire reductive glutamine metabolism pathway, involving several enzymes (glutaminase, glutamate dehydrogenase, isocitrate dehydrogenase, and aconitase) and metabolic co-factors. For example, considering our finding that reductive glutamine metabolism in SDH-KO cells is catalyzed by the mitochondrial NADPH-dependent reduction of alpha-ketoglutarate to isocitrate/citrate by IDH2, a source of mitochondrial NADPH would be needed while supplying NADPH from an exogenous source is problematic (as NADPH cannot cross the mitochondrial membrane). Indeed, conducting a similar experiment to the one described in reply to Comment #2, we detected no citrate m+5 when feeding isolated mitochondria from SDH-KO cells with a physiological concentration of 0.5 mM [U-¹³C]-glutamine (while adding 40 μM NADPH). In fact, while reductive glutamine metabolism was demonstrated time-and-again in literature via isotopic glutamine

tracing on a whole-cell level⁴⁻⁶, we could not find reports of this metabolic route working in isolated mitochondria (and specifically in mitochondria impaired cells such as the one utilized here). This, of course, does not contradict our claim regarding reverse CS flux in SDH-KO mitochondria - which is now further validated with cell lysates, isolated mitochondria, and with chemical and genetic-based inhibition of ACLY (also see below) – it only strengthens the importance of our approach for probing mitochondrial metabolic flux under physiological conditions.

Comment 4: *The authors should silence ACLY and show that M1 malate fractions are not affected when using a 1-13C1 glutamine tracer.*

As suggested, we measured the fractional labeling of malate m+1 in siACLY-transfected cells compared with control cells. siRNA transfections were performed as previously described⁷. Briefly, SDH-KO cells were transfected with Lipofectamine RNAiMAX (Invitrogen) and siRNA pool targeting murine ACLY (Dharmacon #L-040092-01) or a non-targeting control (Dharmacon #D-001810-01-20) at a final concentration of 20 nM (Methods). Western blot analysis of total protein extracts from cells transfected with siControl or siACLY confirmed ACLY was effectively silenced (Fig. X3a). Feeding SDH-KO cells with [1-¹³C]-glutamine for 24hrs, the fractional labeling of citrate m+1 and malate m+1 was not significantly affected by the knockdown of ACLY, suggesting that ACLY contribution to citrate cleavage is insubstantial in these cells (Fig. X3b; lines 295-296).

Fig. X3: Citrate cleavage in SDH-deficient cells is ACLY independent. (a) Immunoblots of proteins in SDH-KO cells transfected with siACLY or siControl. (b) Fractional labeling of citrate (m+1) and malate (m+1) from [1-¹³C]-glutamine in SDH-KO cells transfected with siACLY or siControl.

Comment 5: *It is not clear how mitochondrial volume was determined. The volume is required for calculation of concentrations of mitochondrial metabolites and for Gibbs free energy determination. The volume of mitochondria can vary significantly between conditions and cell lines.*

We agree with the reviewer that mitochondrial volume is required for calculating concentrations of mitochondrial metabolites. However, we do not report metabolite concentrations, but rather metabolite pool sizes in mitochondria and cytosol - and hence did not measure mitochondrial volume. Notably, the Gibbs free energy of IDH1/2/3 was calculated based on the citrate:alpha-ketoglutarate and NAD(P)H:NAD(P)⁺ concentration ratios in the cytosol (IDH1) and mitochondria (IDH2/3); which are equal to the corresponding pool size ratios.

Comment 6: *The title is misleading. While the authors exclusively studied HeLa cells, they generalize these results to cancer cells in common. They should reproduce their results in a few more well characterized cell lines such as A549 or HepG2 to test*

whether this observation is typical for the studied cell model or common to cancer cells.

Our method was applied to study reductive glutamine metabolism in two cell line models: On HeLa cells (under normoxic, hypoxic, and while feeding with standard or physiological glutamine levels); and on SDH positive and negative mouse kidney cells. We do not claim that our finding regarding the importance of IDH1 in normoxia is a common feature in all cell lines; nor do we claim that the reversal of CS in SDH-deficient cells is a common phenotype of all mitochondria impaired cells. The aim of the current study is to establish a new methodology for exploring compartment-specific metabolic flux and demonstrate its applicability in revisiting our basic understanding of metabolic flux in cancer cells – providing insight that is non-observable on a whole-cell level. We expect future research to utilize the spatial-fluxomics method to a variety of cell lines under numerous conditions to explore metabolic flux reprogramming in cancer from a subcellular compartment level.

Comment 7: *It is surprising that the authors found a high ratio of mitochondrial to cytosolic AMP (Fig 1h). I would expect the opposite under normal conditions. Please discuss.*

In accordance with previous studies, mitochondrial AMP level was higher than its cytosolic counterpart^{8,9}.

Comment 8: *There are no measurements for NADP levels. However, the ratio NADPH/NADP is critical for IDH1 and IDH2 activities. It is not clear why the NADH/NAD and NADPH/NADP ratios have been assumed and not measured. These ratios play a key role for some of the main results of the manuscript (e.g. reverse IDH1 flux under normoxia).*

Following the reviewer's comment, we applied our fractionation method to measure the mitochondrial NADH/NAD ratio, utilizing chemical standards to obtain absolute concentrations. We found a mitochondrial NADH:NAD ratio of 0.015 (with a 95% confidence interval of [0.013 0.020]) in HeLa cells under standard normoxic conditions, which is consistent with the assumed ratio of 0.010 ± 0.003 based on previous reports¹⁰⁻¹². The measured NADH/NAD ratios in each cell/conditions are given in Table S2.

NADPH/NADP ratio is highly unstable which complicates its measurement when combined with our fractionation method. Hence, our analysis assumed a wide range of potential NADPH/NADP ratios of ~30-100 in both mitochondria and cytosol, in accordance with multiple reported estimates¹³⁻¹⁵. Notably, while we cannot exclude the possibility of a potential lower/higher mitochondria and/or cytosolic NADPH/NADP ratios, this would have a negligible effect on our main findings. Specifically, the importance of IDH1 as the sole contributor to carbons for citrate production in HeLa cells under normoxia is also evident directly from the isotope tracing data without incorporating thermodynamic constraints in the MFA analysis, identifying a net transport of citrate from the cytosol to mitochondria (i.e. by removing the thermodynamic constraint in Equation 4 from the non-convex optimization). We now relate to the difficulty of performing compartmentalized NADPH/NADP ratio measurements in the Methods (lines 577-586).

Comment 9: *Maybe I missed, how acetyl-CoA was measured. The absolute amount of mitochondrial acetyl-CoA is important for the claimed reverse flux through CS.*

We agree with the reviewer that the absolute quantification of mitochondrial metabolites involved in the reverse CS reaction can provide additional proof to our claim. However, thermodynamics analysis requires the absolute concentrations of all substrates and products involved in the reaction (citrate, CoA, acetyl-CoA, and OAA). Unfortunately, OAA is unstable and not measurable with LC-MS-based metabolomics approaches. Hence, the reversibility of CS was not proven based on thermodynamics analysis of CS, but rather based on the MFA analysis of isotope tracing data. Specifically, the high rate of mitochondrial citrate production via reductive IDH flux is found to flow towards reversed CS due to mass-balance considerations (as a potential citrate transport from mitochondria to cytosol is low). The reversal of CS is also directly evident by the faster labeling of mitochondrial malate m+3 than cytosolic citrate m+5 when feeding [U-¹³C]-glutamine (see Figure 5b). Additional experiments suggested by the reviewer further support the reversal of CS flux (see reply to Comments #1-4).

Comment 10: *Alanine seems to be low in mitochondria vs cytosol. However, previous publications show the opposite (e.g. Buescher et al 2015). Please discuss.*

Buescher et al. claimed that alanine is produced extensively from mitochondrial pyruvate and therefore alanine labeling better reflects the labeling pattern of mitochondrial pyruvate. However, this does not entail that the mitochondrial alanine pool or concentration is higher in mitochondria. In fact, David Sabatini's lab recently reported that the mitochondrial alanine concentration is significantly lower (~10-fold) than that in the whole cell¹², which is consistent with our observation.

Comment 11: *It is not clear how quantitative uptake and secretion fluxes have been measured.*

We did not measure metabolite uptake or secretion rates, as our non-stationary based MFA approach for inferring fluxes does not rely on such data (it rather relies on measuring the absolute concentration of metabolites and metabolite isotopic labeling kinetics; see Compartmentalized metabolic flux analysis (MFA) in Methods).

Responses to reviewer 2's comments:

Comment 1: *I think it should be clearly stated (also in the abstract) that a part of the analyses was made using human HeLa and another one using SDH-deficient mouse cells. Arguments supporting and justifying the comparison between the two species should be added.*

Following the reviewer's comment, we now explicitly state in the Abstract that metabolism in mitochondria impaired cells was studied using SDH-deficient mouse cells. Our goal was to analyze reductive glutamine metabolism under hypoxia and in cells with defective mitochondria, considering previous claims regarding the importance of reductive carboxylation in these cases (and not to compare human/mouse cellular metabolism). Towards this end, we utilized a commonly studied human cell line, HeLa, under normoxic and hypoxic conditions; and an immortalized mouse kidney cell line with impaired mitochondria due to a loss-of-function mutation of the tumor suppressor SDH, which was generated and used before as a model system for studying human SDH-deficient cancers¹⁶.

Comment 2: *The authors investigate carefully the errors introduced during sampling, quenching and metabolite analysis in the two compartments. Most of this work is summarized in Fig. 1. They show clearly that the mitochondria remain intact (compartment specific staining of TMRM and of citrate synthase). This does, however, not strictly prove that all other metabolites remain in the mitochondria to the same extent. They show that the fraction of metabolites where the levels of the sum of subcellular metabolites agree fully with whole cell levels is about 60% (Fig. 1e). About 90% of the metabolite levels agree to about 15% and better. Here it would be important to specify the positioning of the metabolites that are of central interest in this study, e.g. glutamine, citrate, malate, etc. Was a deconvolution also carried out for individual metabolites beyond those shown in Supplementary Table 2?*

For all key metabolites whose compartmentalized measurements were used for flux analysis, the sum of mitochondrial and cytosolic pool sizes closely matched to that of the whole-cell measurement. For malate and citrate, the sum of mitochondrial and cytosolic pool sizes perfectly matched to that measured on a whole cell level; and for alpha-ketoglutarate, the sum of pool sizes deviated less than 13% of the whole-cell measurement (now specified in lines 104-106). Deconvolution of metabolite pool size measurements in the mitochondrial and cytosolic fraction was performed for all metabolites in Figure 1h (data shown in Table 1).

Comment 3: *Similarly, the dynamics of these individual metabolites (Fig. 1.f) would be of interest. How are the time constants related to the metabolic time constants (=flux/concentration) of the network part analyzed in detail? What impact does this have on the flux analysis?*

Showing the drop in mitochondrial metabolite pool sizes when delaying the metabolite extraction was aimed to emphasize the need of rapid fractionation and quenching of metabolism to obtain a reliable view of physiological concentrations (the method by Chen et al., 2016 requires ~15 minutes).

Additional one-minute delay before metabolite quenching leads to a 2-13% drop in malate, citrate, and alpha-ketoglutarate concentrations (Figure X4). The drop in the concentration of these metabolites within 25 seconds before mitochondrial metabolism is quenched with our approach is hence expected to be ~1-4% - which would have a negligible effect on our results.

Fig. X4: Relative pool sizes of alpha-ketoglutarate, citrate, and malate upon delaying the quenching of metabolism in the mitochondrial fraction.

Comment 4: The authors were obviously also looking at extracellular metabolites (see Fig. 3e and l. 167). Did the secretion and uptake of these or other metabolites have any influence on the dynamic labelling data? Ideally, show such data in an supplementary tables!

We analyzed the labeling pattern of secreted citrate and malate in media after 24 hours feeding with [U-¹³C]-glutamine to validate the measured cytosolic labeling patterns with our fractionation method (as shown in Fig 3e). The labeling of media citrate within 3 hours feeding with isotopic glutamine, which is the time interval used for non-stationary MFA analysis of metabolic fluxes, was negligible and below the level of detection by LC-MS. Accounting for a potential uptake of citrate within the 3 hours interval had no effect on the estimated fluxes (the MFA analysis was focused on modeling the isotopic labeling of cytosolic/mitochondrial citrate based on the measured compartmentalized labeling kinetics of malate and alpha-ketoglutarate).

Comment 5: The potential fate of the OAA synthesized in the mitochondria of SDH-KO cells is discussed extensively. What could happen with the mitochondrial acetyl-CoA in this case, when the citrate synthase is running in the opposite direction?

Mitochondrial acetyl-CoA can be converted to acetate and exported out to cytosol¹⁷, acetylate mitochondrial metabolites and proteins¹⁸, participate in amino acid metabolism¹⁹, or be employed to synthesize ketone bodies (acetoacetate, acetone, and β -hydroxybutyrate)²⁰. We found a significant increase in the m+2 labeling of acetylated amino acids (N-acetyl-glutamine, O-acetyl-serine, and N-acetyl-aspartate) when feeding the SDH-KO cells with [U-¹³C]-glutamine versus in SDK-WT cells (Fig. X5a). Consistently, we found that SDH-KO mitochondria have a ~2.3-fold larger acetoacetate pool and a ~1.7-fold increased acetoacetate m+4 (when fed with [U-¹³C]-glutamine; acetoacetate is synthesized by the condensation of two acetyl-CoA derived acetyl groups) compared to that of the wild type (Fig. X5b-c). Hence, ketogenesis, triggered by citrate cleavage in SDH-KO mitochondria, might contribute to tumor growth and survival (acetoacetate converted to β -hydroxybutyrate via BDH1

exerts an anti-inflammatory response²¹). Exploring whether the induced acetylation and production of acetoacetate play an important role in the viability of SDH-KO cells goes beyond the scope of the current study. The above results and discussion regarding the fate of mitochondrial acetyl-CoA are included in lines 373-385.

Fig. X5: Mitochondrial acetyl-CoA from citrate cleavage participates in amino acid metabolism and ketogenesis. (a) Fractional labeling of N-acetylglutamine, O-acetylserine, and N-acetylaspartate in mitochondria. (b) Acetoacetate pool sizes in SDH-WT and KO mitochondria (**P < 0.01 by two-sample t-test). (c) Fractional labeling of acetoacetate in mitochondria.

Comment 6: (Fig. 1h) A log x-scale would be more informative, particular for the metabolites in the upper part of the diagram.

We revised the figure as suggested.

Comment 7: (L 271) *This is actually a higher reductive flux of alpha-ketoglutarate that was estimated.*

Indeed, the reductive IDH2 flux in mitochondria increases ~10-fold in SDH-defective cells compared to the wild-type cells (resulting in more than 7-fold higher reductive glutamine flux in mitochondria than in cytosol; lines 270-273)

Comment 8: (L 380) *The SDH-WT and SDH-KO cells should clearly refer to the cells supplied by Dr. Gottlieb.*

We clarified the text: “HeLa cells were obtained from ATCC while SDH-WT (Sdhb^{fl/fl}) and homogenous SDH-KO (Sdhb^{Δ/Δ}) cells were the gifts of Prof. Eyal Gottlieb (Technion).” (lines 413-414)

Comment 9: (Supplementary Fig. 1) *I could not find a clear definition of the x-axis (standard). What was exactly used as standard?*

We clarified the text: “Standard curve was generated by loading different volume of the whole-cell sample on the gel (0.8, 5, 10, and 30 μL for CS; 0.8, 5, 20, and 30 μL for GAPDH).” (lines 453-454; Supplementary Fig. 1)

Comment 10: (Supplementary Table 1) *A more detailed description (heading and units of data presented) is needed.*

We added a more detailed description of the data in the supplementary table.

Comment 11: (L 91) *Typo – molecule marker.*

Text corrected.

Comment 12: (Supplementary Fig. 2f) *The CO2 released during the synthesis of Uracil on the right hand side should be unlabeled contrary to the one on the left.*

We revised as suggested.

Having addressed the referee comments, we hope you will find our manuscript as meeting the requirements for publication in *Nature Communications*.

Sincerely yours,

Tomer Shlomi

Reference

1. Mall, A. *et al.* Reversibility of citrate synthase allows autotrophic growth of a thermophilic bacterium. *Science (80-.)*. **359**, 563–567 (2018).
2. Shah, S. *et al.* Targeting ACLY sensitizes castration-resistant prostate cancer cells to AR antagonism by impinging on an ACLY-AMPK-AR feedback mechanism. *Oncotarget* **7**, 43713–43730 (2016).
3. Al Kadhi, O., Melchini, A., Mithen, R. & Saha, S. Development of a LC-MS/MS method for the simultaneous detection of tricarboxylic acid cycle intermediates in a range of biological matrices. *J. Anal. Methods Chem.* **2017**, (2017).
4. Jiang, L. *et al.* Reductive carboxylation supports redox homeostasis during anchorage-independent growth. *Nature* **532**, 255–258 (2016).
5. Metallo, C. M. *et al.* Reductive glutamine metabolism by IDH1 mediates lipogenesis under hypoxia. *Nature* **481**, 380–384 (2012).
6. Mullen, A. R. *et al.* Reductive carboxylation supports growth in tumour cells with defective mitochondria. *Nature* **481**, 385–388 (2012).
7. Wellen, K. E. *et al.* ATP-citrate lyase links cellular metabolism to histone acetylation. *Science (80-.)*. **324**, 1076–1080 (2009).
8. Matuszczyk, J. C., Teleki, A., Pfizenmaier, J. & Takors, R. Compartment-specific metabolomics for CHO reveals that ATP pools in mitochondria are much lower than in cytosol. *Biotechnol. J.* **10**, 1639–1650 (2015).
9. Ferraro, P., Nicolosi, L., Bernardi, P., Reichard, P. & Bianchi, V. Mitochondrial deoxynucleotide pool sizes in mouse liver and evidence for a transport mechanism for thymidine monophosphate. *Proc. Natl. Acad. Sci.* **103**, 18586–18591 (2006).
10. Williamson, D. H., Lund, P. & Krebs, H. a. The redox state of free nicotinamide-adenine dinucleotide in the cytoplasm and mitochondria of rat liver. *Biochem. J.* **103**, 514–527 (1967).
11. Sies, H., Akerboom, T. P. & Tager, J. M. Mitochondrial and cytosolic NADPH systems and isocitrate dehydrogenase indicator metabolites during ureogenesis from ammonia in isolated rat hepatocytes. *Eur J Biochem* **72**, 301–307 (1977).
12. Chen, W. W., Freinkman, E., Wang, T., Birsoy, K. & Sabatini, D. M. Absolute Quantification of Matrix Metabolites Reveals the Dynamics of Mitochondrial Metabolism. *Cell* **166**, 1324–1337.e11 (2016).
13. SIES, H., AKERBOOM, T. P. M. & TAGER, J. M. Mitochondrial and Cytosolic NADPH Systems and Isocitrate Dehydrogenase Indicator Metabolites during Ureogenesis from Ammonia in Isolated Rat Hepatocytes. *Eur. J. Biochem.* **72**, 301–307 (1977).
14. Hedekov, C. J., Capito, K. & Thams, P. Cytosolic ratios of free [NADPH]/[NADP⁺] and [NADH]/[NAD⁺] in mouse pancreatic islets, and

- nutrient-induced insulin secretion. *Biochem. J.* **241**, 161–167 (1987).
15. Veech, R. L., Eggleston, L. V & Krebs, H. a. The redox state of free nicotinamide-adenine dinucleotide phosphate in the cytoplasm of rat liver. *Biochem. J.* **115**, 609–619 (1969).
 16. Cardaci, S. *et al.* Pyruvate carboxylation enables growth of SDH-deficient cells by supporting aspartate biosynthesis. *Nat Cell Biol* **17**, 1317–1326 (2015).
 17. Van Grinsven, K. W. A. *et al.* Acetate:succinate CoA-transferase in the hydrogenosomes of *Trichomonas vaginalis*: Identification and characterization. *J. Biol. Chem.* **283**, 1411–1418 (2008).
 18. Pougovkina, O. *et al.* Mitochondrial protein acetylation is driven by acetyl-CoA from fatty acid oxidation. *Hum. Mol. Genet.* **23**, 3513–3522 (2014).
 19. Mifflin, B. J. & Lea, P. J. Amino acid metabolism. *Annu. Rev. Plant Physiol.* **28**, 299–329 (1977).
 20. Newman, J. C. & Verdin, E. Ketone bodies as signaling metabolites. *Trends in Endocrinology and Metabolism* **25**, 42–52 (2014).
 21. Puchalska, P. & Crawford, P. A. Multi-dimensional Roles of Ketone Bodies in Fuel Metabolism, Signaling, and Therapeutics. *Cell Metabolism* **25**, 262–284 (2017).

REVIEWERS' COMMENTS:

Reviewer #1 (Remarks to the Author):

The authors addressed all my critical points and now provide more evidence on reverse CS activity. The additional experiments were carefully performed and based on the new data a reverse flux through CS seems feasible. This significantly strengthened the authors' hypotheses.

Related to their reply to my point 3: It has indeed been shown that fractionated mitochondria operate major parts of their metabolism. To increase the NADPH/NAD⁺ ratio in mitochondria, an inhibition of complex I could be applied. This will induce reductive carboxylation in mitochondria. However for my opinion, this experiment is not needed in regard of this manuscript anymore, since the authors already demonstrated the reverse CS activity in well controlled experiments starting from citrate.

Reviewer #2 (Remarks to the Author):

I think the additional studies have considerably improved the manuscript. I consider it ready for publication now.

Tomer Shlomi, Associate Prof.
Faculty of Biology and Computer
Science
Technion, Haifa 32000, Israel
Tel: +972-4-829-4356
Fax: +972-4-829-3900

February 11, 2019

Response to reviewer 1's comment:

***Comment 1:** The authors addressed all my critical points and now provide more evidence on reverse CS activity. The additional experiments were carefully performed and based on the new data a reverse flux through CS seems feasible. This significantly strengthened the authors' hypotheses.*

Related to their reply to my point 3: It has indeed been shown that fractionated mitochondria operate major parts of their metabolism. To increase the NADPH/NAD⁺ ratio in mitochondria, an inhibition of complex I could be applied. This will induce reductive carboxylation in mitochondria. However for my opinion, this experiment is not needed in regard of this manuscript anymore, since the authors already demonstrated the reverse CS activity in well controlled experiments starting from citrate.

We thank the reviewer for appreciating the improvements of the revised version of the manuscript.

Response to reviewer 2's comment:

***Comment 1:** I think the additional studies have considerably improved the manuscript. I consider it ready for publication now.*

We thank the reviewer for appreciating the revised version of the manuscript.

Having addressed the referee comments, we hope you will find our manuscript as meeting the requirements for publication in *Nature Communications*.

Sincerely yours,

Won Dong Lee and Tomer Shlomi